# Recurrent deletions in clonal hematopoiesis are driven by microhomology-mediated end joining

Tzah Feldman[1], Akhiad Bercovich[2,12], Yoni Moskovitz[1,12], Noa Chapal-Ilani[1], Amanda Mitchell[3], Jessie J. F. Medeiros [3,4], Tamir Biezuner [1], Nathali Kaushansky[1], Mark D. Minden[3,5,6,7], Vikas Gupta[3,6,7], Michael Milyavsky[8,9], Zvi Livneh[10], Amos Tanay [2] & Liran I. Shlush [1,3,11✉]

The mutational mechanisms underlying recurrent deletions in clonal hematopoiesis are not entirely clear. In the current study we inspect the genomic regions around recurrent deletions in myeloid malignancies, and identify microhomology-based signatures in *CALR*, *ASXL1* and *SRSF2* loci. We demonstrate that these deletions are the result of double stand break repair by a PARP1 dependent microhomology-mediated end joining (MMEJ) pathway. Importantly, we provide evidence that these recurrent deletions originate in pre-leukemic stem cells. While DNA polymerase theta (POLQ) is considered a key component in MMEJ repair, we provide evidence that pre-leukemic MMEJ (preL-MMEJ) deletions can be generated in *POLQ* knockout cells. In contrast, aphidicolin (an inhibitor of replicative polymerases and replication) treatment resulted in a significant reduction in preL-MMEJ. Altogether, our data indicate an association between POLQ independent MMEJ and clonal hematopoiesis and elucidate mutational mechanisms involved in the very first steps of leukemia evolution.

[1] Department of Immunology, Weizmann Institute of Science, Rehovot, Israel. [2] Department of Computer Science and Applied Mathematics, Weizmann Institute of Science, Rehovot, Israel. [3] Princess Margaret Cancer Centre, University Health Network (UHN), Toronto, ON, Canada. [4] Department of Molecular Genetics, University of Toronto, Toronto, ON, Canada. [5] Department of Medical Biophysics, University of Toronto, Toronto, ON, Canada. [6] Department of Medicine, University of Toronto, Toronto, ON, Canada. [7] Division of Medical Oncology and Hematology, University Health Network, Toronto, ON, Canada. [8] Department of Pathology, Tel-Aviv University, Tel-Aviv, Israel. [9] Sackler Faculty of Medicine, Tel-Aviv University, Tel-Aviv, Israel. [10] Department of Biomolecular Sciences, Weizmann Institute of Science, Rehovot, Israel. [11] Division of Hematology, Rambam Healthcare Campus, Haifa, Israel. [12] These authors contributed equally: Akhiad Bercovich, Yoni Moskovitz. ✉email: liranshlush3@gmail.com

Human aged hematopoietic stem and progenitor cells (HSPCs) are prone to clonal expansion due to the acquisition of recurrent somatic mutations[1,2]. This phenomenon is known as age related clonal hematopoiesis (ARCH)[3–5]. Somatic pre-leukemic mutations (pLMs) do not usually spread randomly across the possible physical positions of a gene, but rather occur at apparent mutational hotspots. The majority of pLMs are nonsynonymous single nucleotide variants (SNVs), however other pLMs are due to recurrent insertions or deletions (indels)[3]. While the mechanistic explanation for SNVs in cancer has been studied[6], the mechanisms leading to recurrent indels in cancer are less understood. While different indel signatures were previously identified in cancer genomes[6], only two main mutational processes for somatic indels in cancer are mechanistically characterized. The first of which is polymerase slippage, that frequently occurs in repetitive elements and long repeats (microsatellite (MS) signature)[7], while the second is by the error prone process of double strand break (DSB) DNA repair[8,9].

In a recently published study[6], mutation signatures from 4,645 whole-genome and 19,184 exome sequences from different tumor types were analyzed. While 97% of all indels identified in hypermutated cancer genomes carried MS indel signatures in thymine mononucleotide repeats, signatures associated with defective DSB repair were less abundant and mainly reported in BRCA-related tumors (ovarian, breast and cervical carcinomas) owing to deficiencies in the homologous recombination pathway. While other specific indel signatures are associated with tobacco smoking, exposure to UV light and aging, the exact mechanisms underlying them remain to be elucidated.

The study of ARCH and pre-leukemia has been mainly focused on the phenotypic consequences of pLMs, whereas the mutational processes underlying indels signatures in myeloid malignancies and pre-leukemia remain poorly understood. The current study sought to identify deletion signatures in myeloid malignancies that would shed light on the origins and mutational processes promoting these variants.

In this work we demonstrate that the most common recurrent deletions in clonal hematopoiesis are the result of PARP1 dependent repair of DSBs by a sub-pathway of the MMEJ that is POLQ independent.

## Results

### Most common deletions in myeloid malignancies share an MH-based signature.
To study deletion signatures in myeloid malignancies we analyzed targeted sequencing data (COSMIC). This analysis revealed that the most common somatic deletions in myeloid malignancies share a similar signature (Fig. 1a) in which two pre-existing identical sequences (e.g. microhomologies (MHs)) are flanking the deletions (Fig. 1b). The most common somatic MH-based deletions were found in CALR, ASXL1 and SRSF2 genes (Fig. 1a). We validated these results by analyzing deletion signatures in a well-defined targeted sequencing cohort of 1540 adult-AML samples[10]. In this cohort MH-based deletions in ASXL1 and SRSF2 were the most recurrent deletions in AML (Fig. 1c). In a sequencing Myeloproliferative neoplasms (MPN) cohort[11] of 2035 patients, the authors identified by PCR a CALR MH-based 52 bp deletion in 16.2% of 1321 Essential thrombocytosis (ET) patients and 13.6% of 309 Myelofibrosis (MF) patients. This validates that CALR MH-based 52 bp deletion is the most commonly reported deletion in these two clinical entities. Importantly, analysis of this cohort identified a recurrent MH-based deletion in the NFE-2 gene (Fig. 1d). An analysis of an unbiased whole-exome sequencing dataset derived from 562 AMLs (BeatAML)[12] validated the high occurrence rates of ASXL1

and SRSF2 MH-based deletions, and exposed additional non-recurrent MH-based deletions in TET2, DNMT3a, CEBPA and RUNX1 genes (Supplementary Fig. 1a). Taken together, our analyses suggest that the most common deletions in myeloid malignancies are MH-based deletions. We hypothesized that specific mutational mechanisms may underlie these recurrent deletions together with selective pressures. To elucidate the interplay between selective pressures and specific mutational mechanisms, we analyzed somatic mutations reported in the ASXL1 gene.

### High recurrence rates of ASXL1 MH-based deletion in myeloid malignancies are driven by specific mutational mechanisms.
Truncating events occur across the entire exon 12 of the ASXL1 gene and have been suggested to have a gain-of-function role in promoting myeloid malignancies[13]. However, three truncating events were significantly more abundant than others (Supplementary Fig. 2): the nonsense mutation p.R693*, the insertion p.G646fs*12 and the MH-based deletion p.E635fs*15. We compared the frequencies of these common events between hematological and solid tumors. Significant differences were observed in the prevalence of the MH-based deletion (p.E635fs*15) between hematological (153/376 deletion cases) and non-hematological (4/103 cases) cancers ($P < 0.00001$) (Fig. 2a, b). While the MH-based deletion leads to a similar truncated ASXL1 protein as the other two variants (650–700 amino acids), we did not observe similar differences in the frequencies of neither p.R693* nor p.G646fs*12. While we cannot rule out a selective advantage for the MH-based deletion truncated ASXL1 protein specifically in the hematopoietic system, a possible interpretation of these results is that specific mutational mechanisms contribute to leukemogenesis in the myeloid malignancies' cell of origin. To address this hypothesis, we first aimed at identifying the recurrent MH-based deletions' cell of origin.

### Multipotent HSCs are the cells of origin of somatic MH-based deletions.
Somatic mutations in CALR, ASXL1 and SRSF2 genes have been shown by others to be pre-leukemic lesions originating in early multipotent hematopoietic stem cells[14,15]. We wished to validate that multipotent HSCs are the cells of origin for the three recurrent MH-based deletions in these genes. To this end, we first analyzed published sequencing data from healthy individuals and pre-AML cases[14]. We identified the recurrent MH-based deletion in ASXL1 in three of 124 pre-AML cases 10.7, 8.8 and 1.7 years prior to AML diagnoses (Supplementary Table 1) and none among the 676 controls[14]. Additionally, we purified T-cells derived from five AML samples harboring MH-based deletions in ASXL1 and SRSF2 genes (Supplementary Table 2). Somatic MH-based deletions were identified by next-generation sequencing (NGS) in paired T-cells at low allele frequencies (Fig. 3a, Supplementary Table 3). Similarly, we identified the recurrent MH-based CALR deletion in isolated HSPCs and mature cells from two different Myelofibrosis (MF) cases (Supplementary Table 2). This deletion was identified among HSCs, more committed progenitors, and mature myeloid and lymphoid cells (Fig. 3b). We further transplanted CD34 positive cells from one of the cases into NOD/SCID/IL-2Rgc-null (NSG) mice. After 16 weeks multi-lineage graft was observed with the CALR deletion being found in both myeloid and lymphoid cells (Fig. 3c). Taken together, these data highlight the fact that recurrent MH-based deletions in ASXL1, SRSF2 and CALR genes originate in early multipotent HSCs and are part of clonal hematopoiesis. As we provide evidence that these deletions may be the result of specific mutational mechanisms, we aimed to gain insights into these mechanisms by modeling the generation of recurrent MH-based deletions.

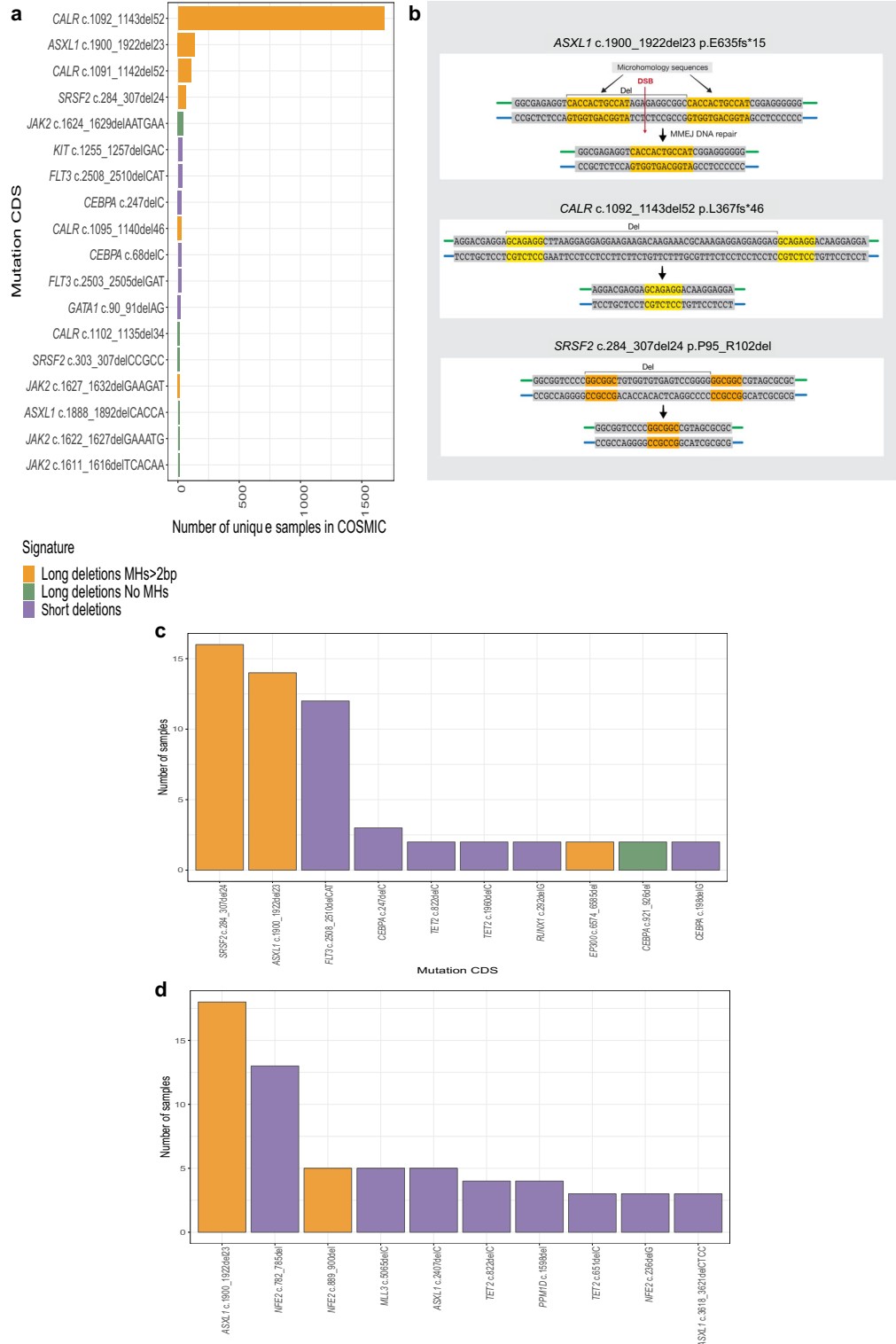

**Fig. 1 Most common deletions in myeloid malignancies share an MH-based signature. a** Number of samples carrying somatic deletion (represented by gene and mutation CDS (coding DNA sequence) names) in myeloid malignancies from COSMIC dataset. Deletions that were identified in 10 or more unique samples are shown. Deletion signatures are: long (≥5-bp) deletions with flanking microhomologies (MHs) of at least 2 bp (orange), long (≥5-bp) deletions with flanking MHs of zero or 1 bp (green) and short deletions (<5-bp) (purple). A single base mismatch was allowed. **b** MH-based deletion signature in *ASXL1*, *CALR* and *SRSF2* genes. Upon double-strand break (DSB) (red arrow) at genomic loci located between two MH sequences (orange and yellow), DNA repair (vertical black arrow) involves a deletion (Del) of one MH and the sequence between the two MHs. **c, d** Number of samples carrying the 10 most common somatic deletions (represented by gene and mutation CDS (coding DNA sequence) names) in 1540 adult acute myeloid leukemia (**c**) and 2045 Myeloproliferative neoplasm (**d**) datasets. Deletion signatures are as described in a. Source data are provided as a Source Data file.

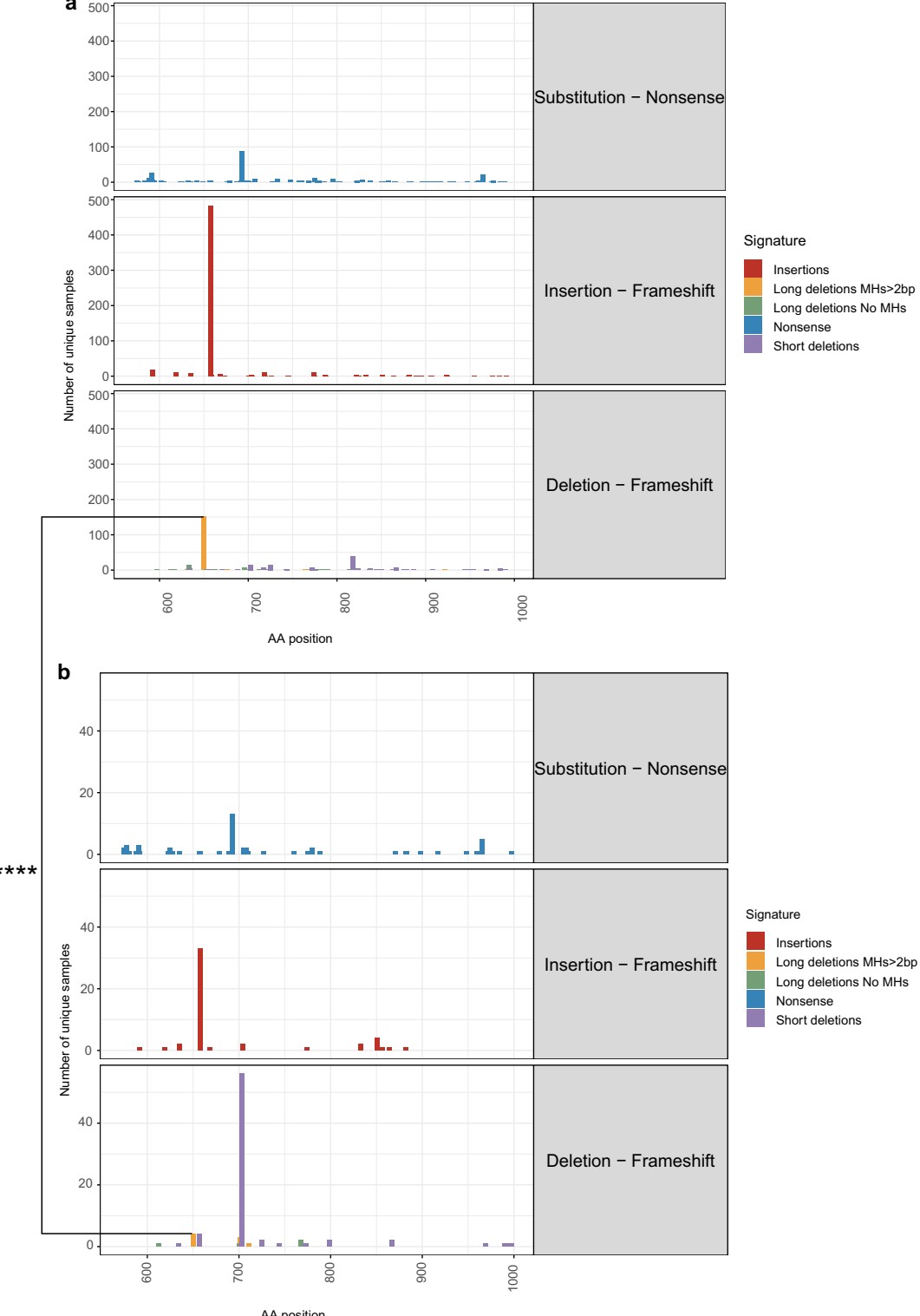

**Fig. 2 High recurrence rates of *ASXL1* MH-based deletion in myeloid malignancies are driven by specific mutational mechanisms. a, b** Number of samples carrying somatic truncating mutations in *ASXL1* gene and the position of the last amino acid of ASXL1 protein (AA position) as identified across $n = 1434$ hematologic (**a**) and $n = 252$ non-hematologic (**b**) tumors in COSMIC dataset. Deletions signatures are Nonsense substitutions (blue), frameshift insertions (red), frameshift ≥5-bp deletions with flanking microhomologies (MHs) of at least 2 bp (orange), frameshift ≥5-bp deletions with flanking MHs of zero or 1 bp (green) and frameshift short deletions (<5-bp) (purple). The proportions of MH-based deletion cases (orange) out of the total deletion cases (purple, green and orange) were compared between hematologic (153/376) and non-hematologic (4/103) tumors. Chi squared test with yates correction for continuity was used to determine statistical significance. (****$p = 4.136e-12$). Source data are provided as a Source Data file.

**a**

| Patient no. | SRSF2 c.284_307del24 | | ASXL1 c.1900_1922del23 | |
|---|---|---|---|---|
| | AML | T cells | AML | T cells |
| 1 | | | | |
| 2 | | | | |
| 3 | | | | |
| 4 | | | | |
| 5 | | | | |

**b**

| Population | CALR c.1092_1143del52 | |
|---|---|---|
| | 140122 | 140681 |
| HSC/MPP | | |
| MLP | | |
| CMP/MEP | | |
| GMP | | |
| TC | | |
| BC | | |
| NK | | |
| CD33+ | | |

**c**

| Mouse # | CALR c.1092_1143del52 | |
|---|---|---|
| | Population | 140122 |
| M1_16W | CD33+_Myeloid | |
| M2_16W | CD19+_BC | |
| M2_16W | CD33+_Myeloid | |
| M3_16W | CD19+_BC | |
| M3_16W | CD33+_Myeloid | |
| M4_16W | CD19+_BC | |
| M4_16W | CD33+_Myeloid | |
| M5_16W | CD19+_BC | |
| M5_16W | CD33+_Myeloid | |

Legend:
- 0 — 100
- No DNA
- No mutation
- Mutated 0–0.1%
- 5%
- 25%
- 50%

**Fig. 3 Recurrent MH-based deletions originate in multipotent HSCs.**
**a** Variant allele frequencies (VAFs) (%) of the recurrent *ASXL1* and *SRSF2* MH-based deletions as assessed by deep targeted sequencing (read depth 5000X) in AML blasts and paired T-cells from the peripheral blood of five AML patients. **b**, **c** Variant allele frequencies (VAFs) (%) of the recurrent *CALR* MH-based deletion as detected by droplet digital PCR (ddPCR) in various cell populations sorted directly from peripheral blood of two myelofibrosis (MF) patients (samples 140122 and 140681) (**b**), or in myeloid (CD33 + ) and B-cells (BC, CD19 + ) isolated from xenografts generated in NSG mice 16 weeks post intra-femoral transplantation of sample 140122 (**c**). HSC/MPP, haematopoietic stem cell/multipotent progenitor; MLP, multi-lymphoid progenitor; CMP/MEP, common myeloid progenitor/megakaryocyte erythroid progenitor; GMP, granulocyte monocyte progenitor; TC, T cell; BC, B cell; NK, Natural killer cell. Gray squares indicate populations without sufficient DNA amounts for variant detection; blue bars indicate VAF.

**CRISPR/Cas9 mediated DSBs recapitulate recurrent MH-based deletions in myeloid malignancies.** Since MH-based deletion signatures are considered to be the result of mutagenic DSB repair[8], we studied the generation of these recurrent deletions in vitro by introducing DSBs using the CRISPR/Cas9 system. We introduced sequential DSBs along the hotspot regions of the *CALR*, *ASXL1* and *SRSF2* genes in K562 CML cell line. Specific DSBs successfully recapitulated recurrent MH-based deletions in *ASXL1* and *SRSF2* genes in K562 cells (Fig. 4a, b). However, we

were unable to recapitulate the *CALR* recurrent frameshift MH-based deletion in vitro at high allele frequency (Supplementary Fig. 3). We validated these results by introducing specific DSBs in four different hematologic cell lines of different genomic and cytogenetic backgrounds. Recurrent MH-based deletions in *ASXL1* and *SRSF2* were successfully obtained in all of these cells (Fig. 4c, d, Supplementary Fig. 4). We further introduced these DSBs in primary human CD34 + HSPCs isolated from six individuals between 30 and 63 years of age (Supplementary Table 4). Remarkably, high frequencies of recurrent MH-based deletions in *ASXL1* and *SRSF2* genes were obtained in all six primary samples (Fig. 4e, f, Supplementary Fig. 4). Our CRISPR/Cas9 experiments of sequential DSBs along the *ASXL1* and *SRSF2* genes provide evidence that the relative frequencies of the different deletions (including the recurrent MH-based deletions) are dependent on specific DSB positions (Fig. 4a, b). As we hypothesized that specific DSB repair pathways would also contribute to the obtained indel landscape, we next aimed to understand which repair machinery is involved in generating the recurrent MH-based deletions.

**Recurrent MH-based deletions in myeloid malignancies are the result of PARP1 mediated MMEJ repair.** We demonstrated that specific DSBs in *ASXL1* and *SRSF2* genes lead to similar indel distributions across many cell lines and primary cells of different genetic backgrounds. We therefore continued to study the DSB repair machineries leading to MH-based deletions in K562 cells. To address this, we manipulated key players in the MMEJ and the classical non-homologous end-joining (c-NHEJ) mutagenic DSB repair pathways. In vitro inhibition of the MMEJ pathway by PARP1 inhibitor (Rucaparib camsylate) prior to DSB induction, similar to a previously descried method[16], resulted in significantly reduced allele fractions of both *ASXL1* and *SRSF2* recurrent MH-based deletions (Fig. 5a–f, Supplementary Fig. 5). This provides evidence that these deletions are the result of PARP1 mediated MMEJ repair. To further validate our results, we inhibited the c-NHEJ pathway by generating K562 *LIG4* −/− cells. *LIG4* knockout resulted in a significant increase in the allele fractions of the recurrent MH-based deletions, further ruling out the role of c-NHEJ in generating these variants (Fig. 5c, f, Supplementary Fig. 5). Interestingly, *LIG4* −/− cells did not produce short deletions and insertions close to the breakpoint, suggesting that these indels are due to LIG4 mediated c-NHEJ repair. Moreover, high dosages of rucaparib treatment reduced insertions at the breakpoints, while short deletions were obtained. This indicates that PARP1 may not be specific to MMEJ and it may have some role in c-NHEJ repair, as was previously described[17]. Altogether, our results suggest that recurrent MH-based deletions in *ASXL1* and *SRSF2* are the consequence of PARP1 mediated and LIG4 independent MMEJ repair. We thereafter refer to these deletions as pre-leukemic MMEJ deletions (preL-MMEJ deletions). Of note, while *CALR* recurrent MH-based deletion could be obtained at very low allele fractions in Wild type (WT) K562 cells, a 20-fold increase in allele fraction was observed among *LIG4* −/− cells (Supplementary Fig. 6). DNA Polymerase theta (POLQ) was shown to be a key participant in MMEJ[8,18,19] and also to play a role in CRISPR/Cas9 mediated MMEJ repair[20]. We therefore aimed to assess POLQ contribution to the preL-MMEJ deletions.

**preL-MMEJ deletions can be obtained in *POLQ* knockout cells.** We generated three distinct K562 *POLQ* −/− cells harboring frameshift mutations at exon 14, 16 and 18 of the *POLQ* gene. Each one of these mutations presumably leads to a premature stop codon upstream or inside the polymerase domain, previously

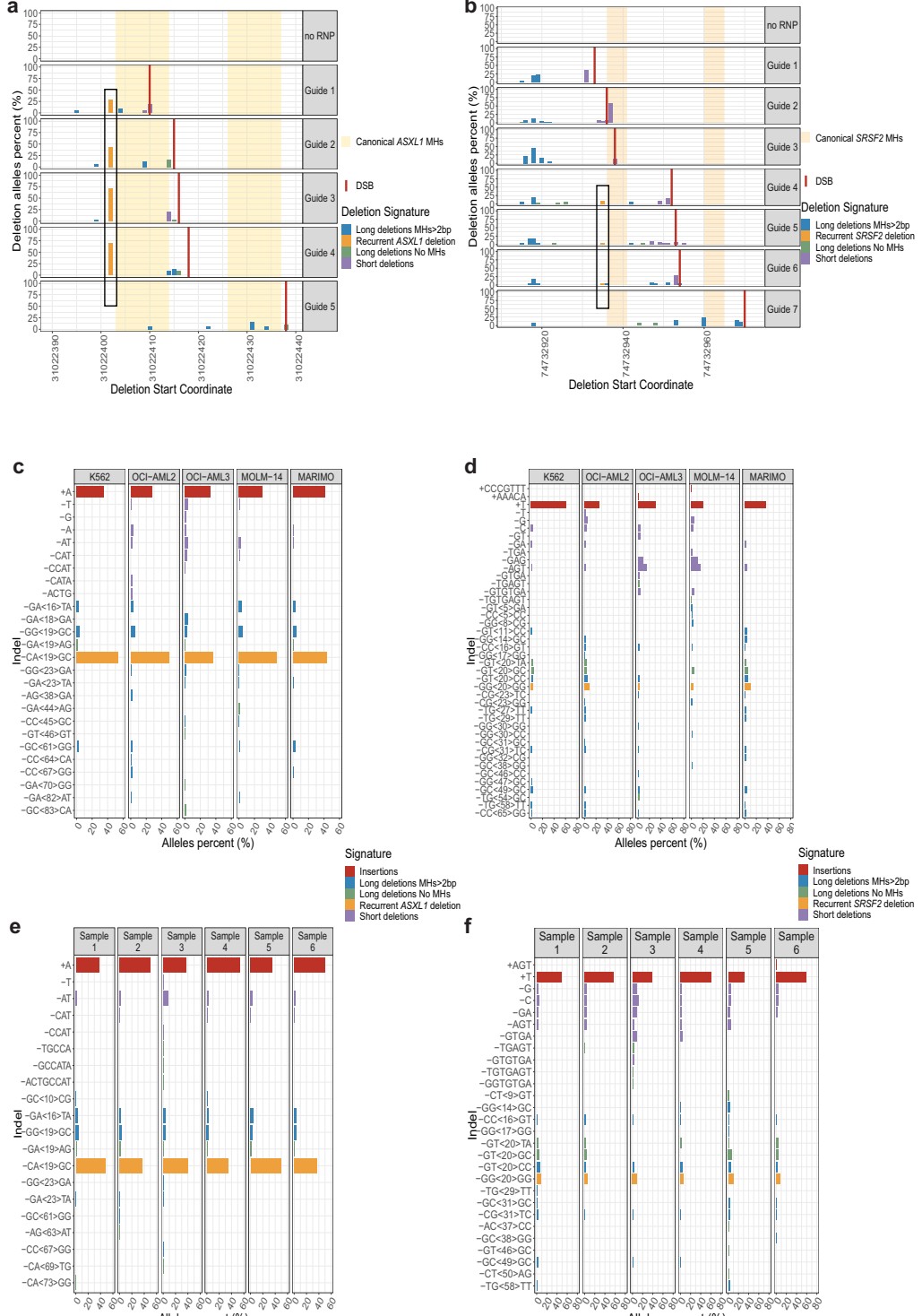

**Fig. 4 CRISPR/Cas9 mediated DSBs recapitulate recurrent MH-based deletions in myeloid malignancies. a, b** Start genomic positions and percentage of *ASXL1* (**a**) and *SRSF2* (**b**) deletion alleles among the total deletion alleles obtained for each sgRNA guide as assessed by deep targeted sequencing (read depth 5000X) in the K562 cell line. Insertions are not shown. Deletion signatures are: ≥5-bp deletion with flanking microhomologies (MHs) of at least 2 bp (blue), ≥5-bp deletion with flanking MHs of zero or 1 bp (green), short deletion (<5-bp) (purple) and the recurrent deletions in *ASXL1* (**a**) or *SRSF2* (**b**) genes (orange). Canonical MHs (orange and yellow backgrounds), sequential DSBs (vertical red lines) and the recurrent deletions (black outlines) are marked along the *ASXL1* (**a**) and *SRSF2* (**b**) sequences. 'no RNP' controls represent samples treated with Cas9 in the absence of sgRNAs. **c–f** Indel sequences and percentage of *ASXL1* (**c, e**) and *SRSF2* (**d, f**) modified alleles among the total indel alleles as assessed by deep targeted sequencing (read depth 5000X) in K562, OCI-AML2, OCI-AML3, MOLM-14 and MARIMO cells (**c, d**) or primary human CD34+ hematopoietic stem and progenitor cells isolated from six individuals (**e, f**) following the induction of DSBs in *ASXL1* (**c, e**) and *SRSF2* (**d, f**) loci. Allele percent of 0.5% (**c, e**) or 1% (**d, f**) and above are shown. Deletion signatures are as described in a, b with the addition of insertions (red). Source data are provided as a Source Data file.

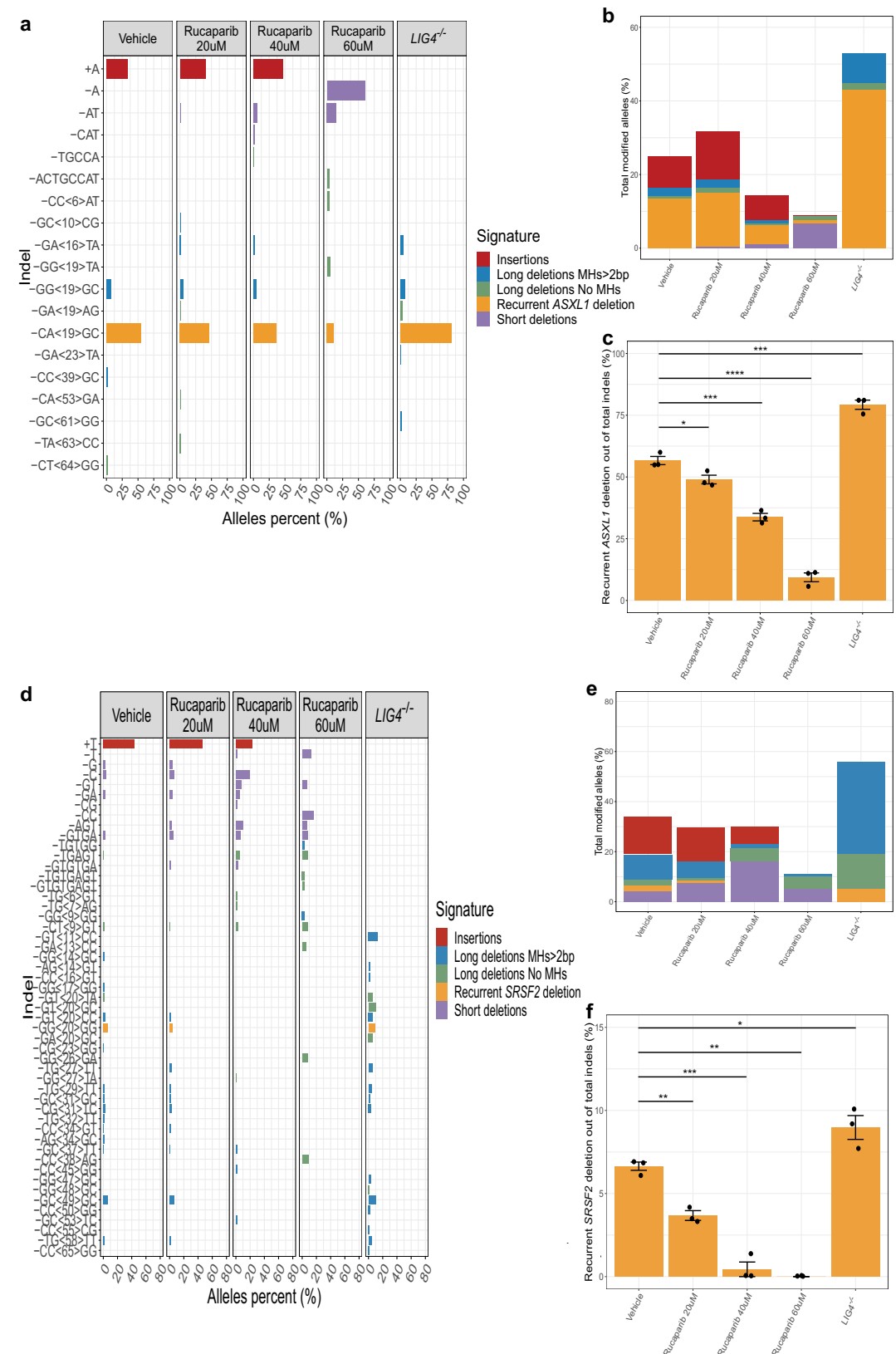

shown to be required for end joining repair[21,22]. In *SRSF2*, a significant decrease in total fractions of MH-based deletions together with an increase in short deletions, validated a role for POLQ in MMEJ (Fig. 6f, g). In contrast, *POLQ* knockout resulted in a mild and mostly insignificant decrease in the fractions of both preL-MMEJ deletions (Fig. 6c, h, Supplementary Fig. 7).

This suggests that POLQ has a limited role in the pathway leading to preL-MMEJ deletions. We therefore hypothesized that other DNA polymerases may collaborate with PARP1 and be involved in the pathway leading to preL-MMEJ deletions in humans. In order to identify such an involvement, we analyzed gene expression data of single human HSCs.

**Fig. 5 Recurrent MH-based deletions in myeloid malignancies are the result of PARP1 mediated MMEJ repair. a, d** Indel sequences and percentage of *ASXL1* (**a**) and *SRSF2* (**d**) modified alleles among the total indel alleles as assessed by deep targeted sequencing (read depth 5000X) in K562 cells following the induction of DSBs in *ASXL1* (**a**) or *SRSF2* (**d**) loci. *LIG4* −/− K562 cells (right panels) are presented together with Wild type (WT) K562 cells that were electroporated in the presence of the DMSO vehicle, 20, 40 or 60 μM rucaparib camsylate as indicated. Allele percent of 1% (**a, d**) and above are shown. **b, e** Overall modification frequency as assessed by deep targeted sequencing (read depth 5000X) in K562 cells following the induction of DSBs in *ASXL1* (**b**) or *SRSF2* (**e**) loci. WT K562 cells that were electroporated in the presence of the DMSO vehicle, 20, 40 or 60 μM rucaparib camsylate, together with *LIG4* −/− K562 cells are shown. **c, f** Percentage of the recurrent MH-based deletions among the total indel alleles following the induction of DSBs in *ASXL1* (**c**) or *SRSF2* (**f**) loci. WT K562 cells that were electroporated in the presence of the DMSO vehicle, 20, 40 or 60 μM rucaparib camsylate, together with *LIG4* −/− K562 cells are shown. Data are presented as mean values ± SEM. $n = 3$ biologically independent samples. Unpaired two tailed T-test was used to determine statistical significance. (*$P < 0.05$, **$P < 0.01$, ***$P < 0.001$, and ****$P < 0.0001$). **c** Vehicle vs. rucaparib 20 μM $p = 0.033$, vehicle vs. rucaparib 40 μM $p = 0.00052$, vehicle vs. rucaparib 60 μM $p = 4.17$ e-05, vehicle vs. *LIG4* −/− $p = 0.00082$. **f** Vehicle vs. rucaparib 20uM $p = 0.0015$, vehicle vs. rucaparib 40 μM $p = 0.00025$, vehicle vs. rucaparib 60 μM $p = 0.0013$, vehicle vs. *LIG4* −/− $p = 0.037$. Indel signatures are: Insertions (red), ≥5-bp deletion with flanking microhomologies (MHs) of at least 2 bp (blue), ≥5-bp deletion with flanking MHs of zero or 1bp (green), short deletion (<5-bp) (purple) and the recurrent deletions in *ASXL1* (**a, b, c**) *SRSF2* (**d, e, f**) genes (orange). Source data are provided as a Source Data file.

**Inhibition of replicative DNA polymerases by aphidicolin reduces the formation of preL-MMEJ deletions.** We next studied the gene expression profiles of human single bone-marrow (BM) progenitor cells as was previously described[23]. We analyzed BM CD34+ profiles from the Human Cell Atlas Consortium's immune census dataset (https://preview.data.humancellatlas.org/) (Supplementary Fig. 8) and focused on multipotent HSCs expressing CD34 and AVP markers, and proliferating MPPs (cells of origin of MH-based deletions) (Supplementary Fig. 9). We noticed that as HSCs enter cell replication, they upregulate components of the c-NHEJ, MMEJ, and HR pathways (Supplementary Fig. 10). Our experimental results demonstrated that inhibition of *PARP1* by rucaparib camsylate resulted in a decreased production of preL-MMEJ deletions in vitro. As we also provide evidence that the preL-MMEJ deletions originate in multipotent HSCs, we assessed for a possible correlation between the expression levels of *PARPs* and a list of human DNA polymerases[24] specifically in HSCs and MPPs, for polymerases that are not correlated throughout all progenitor states. Among this sub-population, *PARP1* expression levels were shown to significantly correlate only in HSCs with *POLQ*, but also with *POLD1*, *POLE* and *POLE4* gene expression levels (Fig. 7a). *POLD1* and *POLE* genes encode for the catalytic subunits of the B-family DNA polymerases delta and epsilon respectively, which are the major replicases that carry out DNA replication in eukaryotes[25]. We next aimed to experimentally assess whether these replicative DNA polymerases contribute to the formation of preL-MMEJ deletions.

To address this issue, we treated K562 WT and *POLQ* −/− cells with low and high doses of aphidicolin, a potent inhibitor of eukaryotic replicative B-family DNA polymerases[26–29]. Remarkably, aphidicolin treatment significantly reduced the fractions of preL-MMEJ deletions in both WT and *POLQ* −/− cells (Fig. 7b–g, Supplementary Fig. 11). The relative contribution of *POLQ* knockout to this reduction seemed to be negligible compared to aphidicolin treatment (Fig. 7b–g). A parallel increase was observed in the fraction of c-NHEJ associated short deletions in both genomic loci (Supplementary Fig. 11b, e). Altogether, these results suggest a sub-pathway of the MMEJ repair, leading to preL-MMEJ deletions. This sub-pathway seemed to be mediated by PARP1, active in the absence of POLQ, and successfully inhibited by aphidicolin treatment.

## Discussion

In the current study, we establish that the three most common pre-leukemic somatic deletions *ASXL1* c.1900_1922del23, *SRSF2* c.284_307del24 and *CALR* c.1092_1143del52 (termed here

preL-MMEJ deletions) share a similar deletion signature suggesting similar underlying mutational processes (Fig. 1). We provide evidence that these hotspot-deletions occur not just due to selective advantage, but also as a result of unique mutational mechanisms (Fig. 2) and that they originate in multipotent HSCs (Fig. 3). All three preL-MMEJ deletions were successfully recapitulated following DSBs (Fig. 4, Supplementary Fig. 6) that are repaired by the PARP1 dependent MMEJ (Fig. 5). Knockout of *POLQ* gene (which encodes the main polymerase involved in MMEJ) did not significantly reduced preL-MMEJ deletions (Fig. 6). Single cell RNA-seq data of human HSCs suggest that the MMEJ pathway is activated as HSCs replicate, and exposed a correlation between the gene expression of *PARP1* and *POLQ*, *POLD1* and *POLE* (Fig. 7a). Finally, inhibition of the replicative polymerases and consequently cellular replication by aphidicolin resulted in a significant reduction of preL-MMEJ deletions (Fig. 7). Collectively, our data provide insights into mutational mechanisms in HSCs and the early stages of clonal hematopoiesis.

In the current study we provide evidence that MMEJ is a major driver in early leukemia evolution. HSCs appear to use MMEJ over c-NHEJ repair as reflected by the much higher prevalence of the preL-MMEJ deletions compared to other short deletions. Our analysis of single cell RNA-seq data together with in vitro experimental results indicate that the MMEJ pathway is active during cell replication. Our findings demonstrate that synchronizing cells at the G1/S boundary by aphidicolin substantially reduced MMEJ, while c-NHEJ repair was relatively increased. This is in line with previous reports demonstrating a significant elevated MMEJ activity during S and G2 cell cycle phases owing to CtIP phosphorylation as cells enter S-phase[30,31]. Phosphorylated CtIP is in turn stimulating the MRN complex mediated end-resection, which is a critical step in the initiation of both MMEJ and HR[32]. However, the full biological scenario in which DSBs occur in HSCs remains unclear and is important to understand in order to potentially prevent preL-MMEJ deletions. One possible scenario as was previously suggested[33] is that different types of physiological stress lead to DNA damage and consequently to the exit of HSCs from dormancy. An alternative scenario is that aged HSCs carry more DSBs and Gamma H2AX foci due to altered dynamics of DNA replication forks[34]. In both cases, MMEJ may be the preferred repair choice as it is available and efficient during cell replication. Future studies should shed more light on the origins of the DNA damage leading to preL-MMEJ, either due to extrinsic physiological stress or age related replicative stress. Furthermore, as the mechanism of MMEJ underlying preL-MMEJ deletions is not fully resolved in the current study, a more accurate description of the sub-pathway responsible for preL-MMEJ deletions is needed.

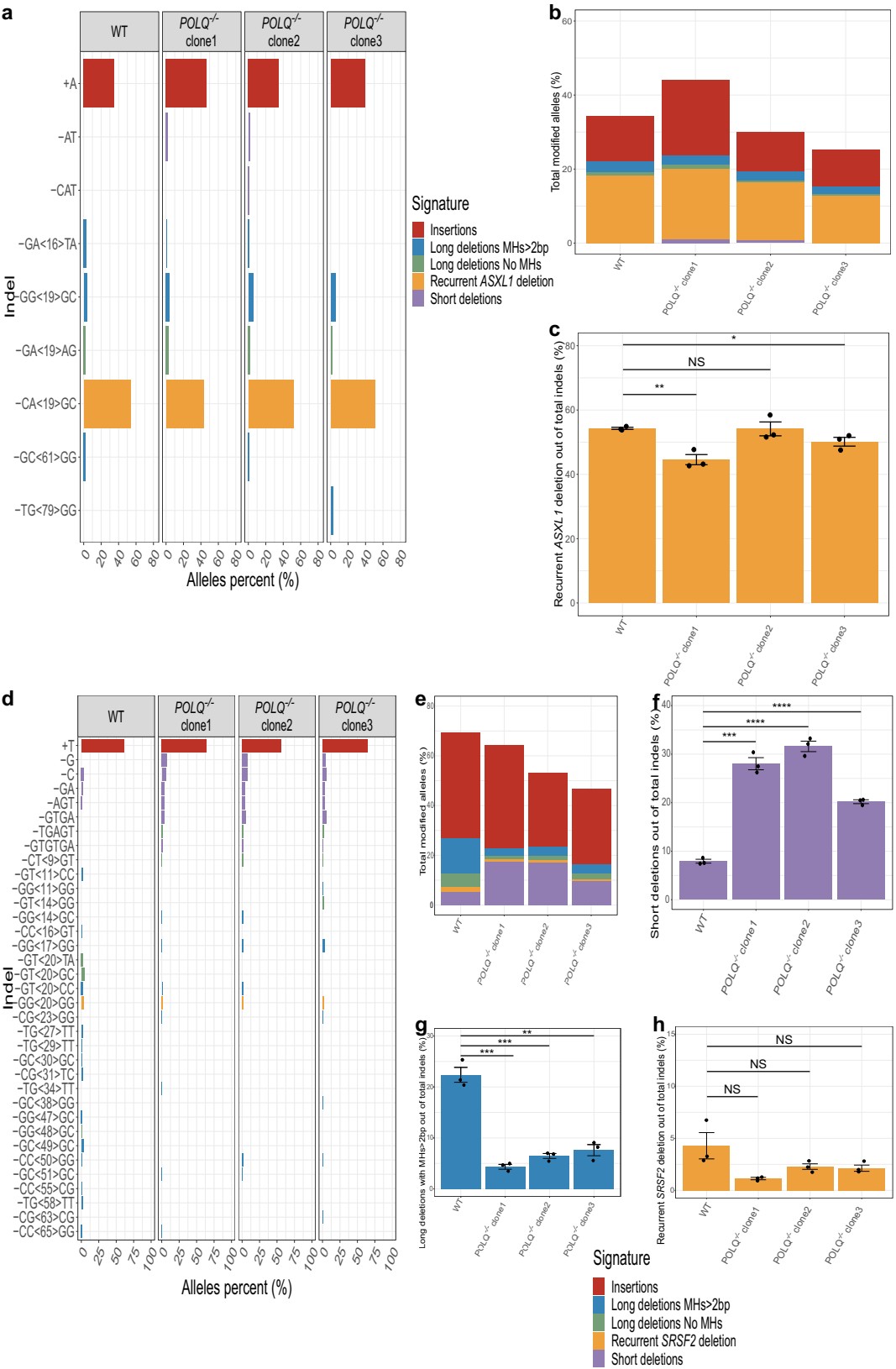

Here we demonstrate that preL-MMEJ deletions are the result of PARP1 mediated and POLQ independent sub-pathway of the MMEJ. While PARP1 is known to regulate the MMEJ pathway[32] it also plays a role in c-NHEJ[17] and single strand break (SSB) repair[35]. We cannot rule out that human preL-MMEJ might be the result of SSB. Strand synthesis during MMEJ should require the involvement of DNA polymerases, we propose a model in which replication associated DNA polymerases are involved in preL-MMEJ (Fig. 8). However, future studies are warranted to assess whether aphidicolin related reduction of preL-MMEJ is due to a direct inhibition of replicative polymerases or as a consequence of cell cycle arrest.

**Fig. 6 preL-MMEJ deletions are obtained in *POLQ* knockout cells. a, d** Indel sequences and percentage of *ASXL1* (**a**) and *SRSF2* (**d**) modified alleles among the total indel alleles as assessed by deep targeted sequencing (read depth 5000X) in K562 cells following the induction of DSBs in *ASXL1* (**a**) or *SRSF2* (**d**) loci. Wild type (WT) K562 together with three distinct clones of *POLQ* −/− K562 cells are shown. **b, e** Overall modification frequency as assessed by deep targeted sequencing (read depth 5000X) in K562 cells following the induction of DSBs in *ASXL1* (**b**) or *SRSF2* (**e**) loci. WT K562 together with three distinct clones of *POLQ* −/− K562 cells are shown. **c, f, g, h** Percentage of the recurrent *ASXL1* MH-based deletions (**c**), <5-bp short deletions in *SRSF2* (**f**), ≥5-bp deletions with flanking microhomologies (MHs) of at least 2 bp in *SRSF2* (**g**) and *SRSF2* recurrent MH-based deletions (**h**) among the total indel alleles following the induction of DSBs in *ASXL1* (**c**) or *SRSF2* (**f, g, h**) loci. WT K562 together with three distinct clones of *POLQ* −/− K562 cells are shown. Data are presented as mean values ± SEM. $n = 3$ biologically independent samples. Unpaired two tailed T-test was used to determine statistical significance. (NS, nonsignificant, *$P < 0.05$, **$P < 0.01$, ***$P < 0.001$, and ****$P < 0.0001$). **c**: WT vs. *POLQ* −/− clone 1 $p = 0.0038$, WT vs. *POLQ* −/− clone 2 $p = 0.94$, WT vs. *POLQ* −/− clone 3 $p = 0.041$. **f** WT vs. *POLQ* −/− clone 1 $p = 0.000105$, WT vs. *POLQ* −/− clone 2 $p = 3.33e-05$, WT vs. *POLQ* −/− clone 3 $p = 2.66e-05$. **g**: WT vs. *POLQ* −/− clone 1 $p = 0.0003$, WT vs. *POLQ* −/− clone 2 $p = 0.0004$, WT vs. *POLQ* −/− clone 3 $p = 0.0012$. **h** WT vs. *POLQ* −/− clone 1 $p = 0.12$, WT vs. *POLQ* −/− clone 2 $p = 0.19$, WT vs. *POLQ* −/− clone 3 $p = 0.17$. Indel signatures are: Insertions (red), ≥5-bp deletion with flanking microhomologies (MHs) of at least 2 bp (blue), ≥5-bp deletion with flanking MHs of zero or 1 bp (green), short deletion (<5-bp) (purple) and the recurrent deletions in *ASXL1* (**a, b, c**) *SRSF2* (**d, e, f, g, h**) genes (orange). Source data are provided as a Source Data file.

PreL-MMEJ deletions are typically identified among the elderly. An important factor contributing to DSB repair choice, might be the age of the cell of origin in which the DSBs occur. In our CRISPR/Cas9 based model, similar frequencies of preL-MMEJ deletions were obtained in young and aged human HSPCs. This might be due to the fact that our model system is not mimicking the exact biological context in which preL-MMEJ arise. It remains unclear whether preL-MMEJ deletions can occur in HSCs at any age and expand due to selective advantage at older age or that preL-MMEJ deletions preferentially occur in aged HSCs. To elucidate this, the phylogenetic origins of preL-MMEJ deletions can be studied in single cells as was previously done[36] to determine the exact age in which they originate.

While preL-MMEJ deletions in *ASXL1* and *SRSF2* are the most recurrent deletions in AML, they are identified in a relatively small proportion of AML patients (~2%). However, these deletions signatures are not the sole hallmark of MMEJ repair. It was recently shown that other genetic alterations such as blunt-end deletions, templated insertions[37] and copy number variations (CNV)[38] were also the result of POLQ mediated MMEJ. Large CNVs containing *DNMT3a* and *TET2* genes can be found among healthy individuals[39], however, the mutational mechanisms promoting them are underexplored. Interestingly, large numbers of AML patients harbor recurrent somatic insertions that are templated from nearby genomic sequences. These duplications can be found along *CALR*, *ASXL1* and *SRSF2* hotspot regions, as well as in *NPM1*, *FLT3* and *MLL* genes. Altogether, it is possible that MMEJ related contribution to genetic alterations in pre-leukemia and leukemia are underestimated.

In the current study, we aimed at understanding the biological processes driving early mutations in myeloid malignancies. Our findings support the growing evidence that some cancer mutations do not occur randomly but rather their physical positions and patterns are determined by more than the selective advantage they provide. DSBs followed by MMEJ repair might shape the mutational landscape observed in myeloid malignancies. In line with this, recent studies demonstrated hyperactivity of the MMEJ pathway in *IDH2*[40] and *FLT3-ITD* mutated AMLs[41] and the sensitivity of some AML[42] and MPN cells[43] to PARP1 inhibition. Such sensitivity could be explained by the dependency of HSPCs on MMEJ and the synthetic lethality of PARP1 inhibitors. Further characterization of these findings is required to potentially intervene with the MMEJ pathway and prevent somatic mutagenesis associated with clonal hematopoiesis.

## Methods

**Samples**. De-identified primary peripheral blood samples were obtained with informed consent from Acute Myeloid Leukemia (AML) and Myelofibrosis (MF) patients through the Leukemia Tissue Bank at Princess Margaret Cancer Centre in accordance with regulated procedures approved by the Research Ethics Board of the University Health Network (REB 01-0573-C). De-identified mobilized peripheral blood autologous transplant products were obtained with informed consent from Non-Hodgkin Lymphoma (NHL), Multiple Myeloma (MM) and Amyloidosis patients through the Leukemia Tissue Bank at Princess Margaret Cancer Centre in accordance with regulated procedures approved by the Research Ethics Board of the University Health Network (ethics committee protocol # 15-9633), and Weizmann institute of science (IRB protocol #337-1). All patients provided written informed consent for the usage of their samples for research purposes and for the usage of their clinical and biological data. We complied with all relevant ethical regulations for work with human participants.

**Primary CD34 + enrichment and pre-electroporation culturing**. CD34 + cells were isolated from mononuclear cells derived from mobilized peripheral blood stem cells (PBSC) autologous transplant products by using EasySep Human CD34 Positive Selection Kit II (StemCell Technologies, 18056). Cell were cultured for 48 h before electroporation in StemSpan™ Serum-Free Expansion Medium II (SFEM II) (StemCell Technologies, 09605) with streptomycin (20 mg/mL), penicillin (20 unit/mL) and the following human cytokines (all from GenScript unless stated otherwise, catalog numbers and dilution used in parentheses): FLT3L (Z02926, 100 ng/mL), G-CSF (Z02980, 10 ng/mL), SCF (Z02692, 100 ng/mL), TPO (Pepro-Tech, 300-18, 25 ng/mL) and IL-6 (Z03034, 10 ng/mL). Cells were cultured at a density of $2.5*10^5$ cells/ml in 96-well U-bottom plates.

**Cell lines pre-electroporation culturing**. K562, Marimo, MOLM-14, OCI-AML-2 and OCI-AML-3 cell lines were used in this study. All cell lines were obtained from ATCC, were authenticated by whole-exome sequencing and tested negative for Mycoplasma contamination. All cell lines were sub-cultured 2 days before electroporation in RPMI 1640 Medium containing L-Glutamine (Biological Industries, 01-100-1 A) with 10% FBS, streptomycin (20 mg/mL) and penicillin (20 unit/mL) at a density of $3*10^5$ cells/ml.

**CRISPR/Cas9 experiments**. 20 bp sgRNA sequences were designed along the genomic loci of interest using DESKGEN algorithm (https://www.deskgen.com/landing/#/login). Sequential DSBs that are described in Fig. 4a, b and Supplementary Fig. 3 were performed using the px330 plasmid system, similar to a previously described method[44,45]. All relevant Px330 plasmid preparations details are described under section 4.1. Electroporation reactions using px330 plasmids were performed at 2 ug purified plasmids per reaction. All other CRISPR/Cas9 experiments were done using sgRNAs guide 2 (*ASXL1*) and 5 (*SRSF2*) that were synthesized from IDT and are detailed under section 4.2.

**px330 plasmid preparations**. For experiments involving sequential DSBs along the *ASXL1*, *SRSF2* and *CALR* sequences, sense and antisense oligonucleotides for each sgRNA with overhangs compatible to Bbsi-digested px330 were designed and ordered from IDT. Each oligos pair was further phosphorylated and annealed using T4 PNK (NEB, M0201S) and T4 Ligation Buffer (NEB, B0202S). Phosphorylation and annealing reactions were performed at 37 °C for 30 min, followed by 95 °C for 5 min and ramping down to 25 °C at 5 °C /min. Annealed oligo pairs were then ligated into a previously Bbsi digested px330 plasmid. Per reaction, 50 ng digested px330 was mixed with 1:250 diluted oligo duplex with 2X quick ligation buffer and quick ligase (NEB, M2200S) at 16 °C overnight.

BioSuper DH5α competent cells (Bio-lab, cat. no. 959758026600) were transformed with Px330. Bacteria was re-suspended and plated on LB agar ampicillin dishes and incubated at 37 °C over-night. Colonies were then screened and grew in 2-3 ml LB + Ampicillin at 37 °C overnight in a shaker (250 rpm). For each colony (sgRNA), plasmid DNA was extracted using the QIAprep Spin Miniprep standard protocol (Qiagen, cat. No. 27104). To validate the presence of the desired inserts,

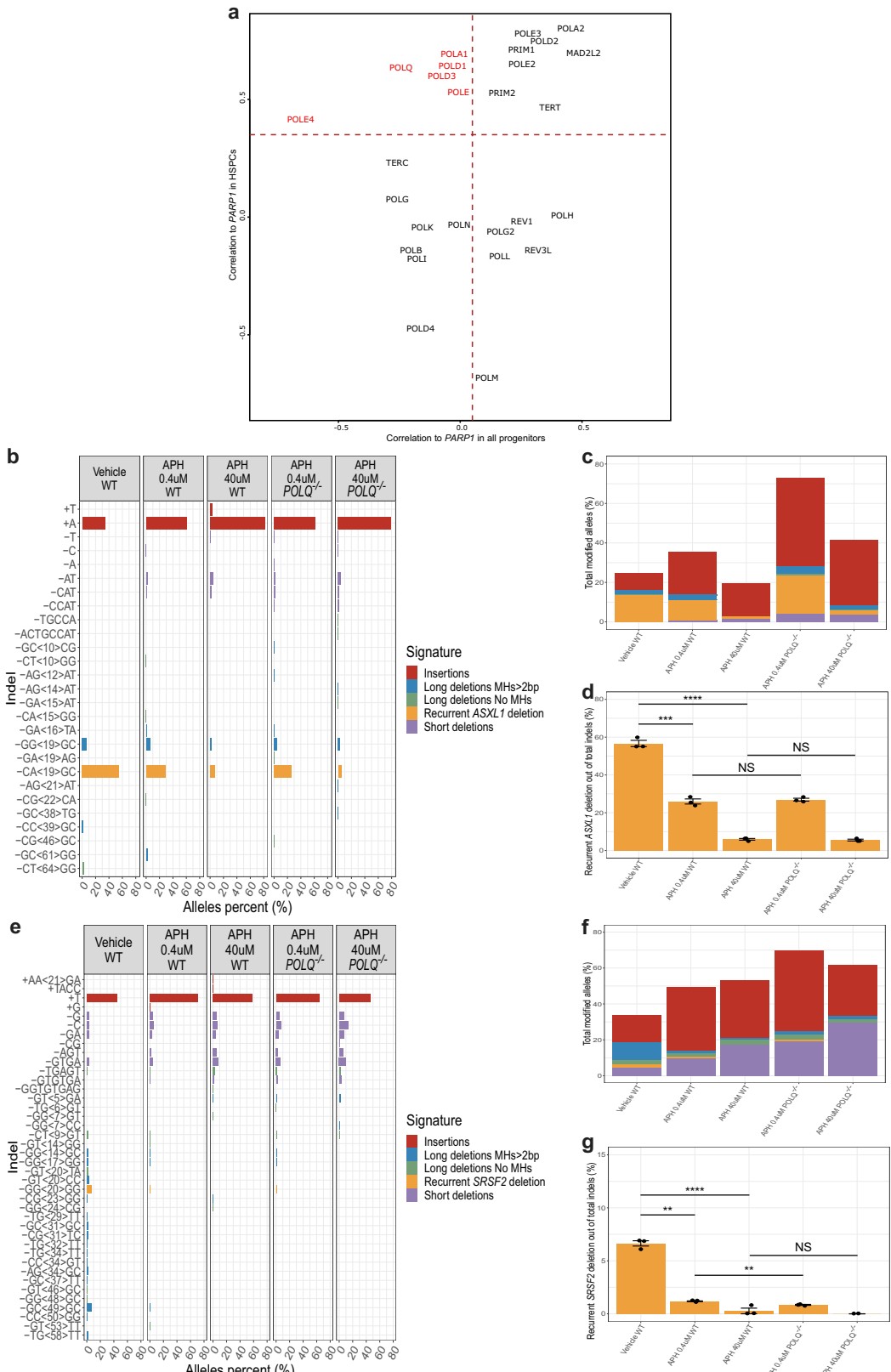

Sanger sequencing reactions were performed for each plasmid using the U6 promoter primer ACTATCATATGCTTACCGTAAC. Electroporation reactions using purified px330 plasmids were done at 2ug plasmid per reaction.

**RNP complex preparations**. All other CRISPR/Cas9 experiments were done using sgRNAs guide 2 (*ASXL1*) and 5 (*SRSF2*) that were synthesized from IDT. Lyophilized sgRNAs were re-suspended in IDTE buffer (PH 7.5) to a final concentration of 100 uM. RNP complex for each reaction were generated by mixing 1.2 ul sgRNA, 1.7 ul Cas9 protein (IDT) and 2.1 ul PBS followed by incubation for 10 min at 20 degrees.

**Electroporation reactions**. All electroporation reactions were performed using the 16-strip Lonza 4D nucleofector kit. Pre-electroporated cells were washed in PBS

**Fig. 7 Aphidicolin treatment reduces the formation of preL-MMEJ deletions. a** Scatter plot of Pearson correlations between *PARP1* and various DNA polymerase genes normalized expression, compared when calculated for all the progenitors metacells in the bone-marrow CD34 + data (X-axis) and specifically for hematopoietic stem and progenitor cells (HSPCs) metacells (Y-axis). Red gene names show DNA polymerases with significant correlation in HSPCs but negligible or negative correlation across all progenitors. **b, e** Indel sequences and percentage of *ASXL1* (**b**) and *SRSF2* (**e**) modified alleles among the total indel alleles as assessed by deep targeted sequencing (read depth 5000X) in K562 cells following the induction of DSBs in *ASXL1* (**b**) or *SRSF2* (**e**) loci. Wild type (WT) or *POLQ* −/− K562 cells that were electroporated in the presence of the DMSO vehicle (only WT), 0.4 or 40 uM aphidicolin (APH) are shown. In **e**, allele percent of 0.5% and above are shown. **c, f** Overall modification frequency as assessed by deep targeted sequencing (read depth 5000X) in K562 cells following the induction of DSBs in *ASXL1* (**c**) or *SRSF2* (**f**) loci. WT or *POLQ* −/− K562 cells that were electroporated in the presence of the DMSO vehicle (only WT), 0.4 or 40 uM aphidicolin (APH) are shown. **d, g** Percentage of the recurrent MH-based deletions among the total indel alleles following the induction of DSBs in *ASXL1* (**d**) or *SRSF2* (**g**) loci. WT or *POLQ* −/− K562 cells that were electroporated in the presence of the DMSO vehicle (only WT), 0.4 or 40 μM aphidicolin (APH) are shown. Data are presented as mean values ± SEM. $n = 3$ biologically independent samples. Unpaired two tailed T-test was used to determine statistical significance. (NS, nonsignificant, **$P < 0.01$, ***$P < 0.001$, and ****$P < 0.0001$). **d** Vehicle WT vs. APH 0.4 μM WT $p = 0.00013$, vehicle WT vs. APH 40 μM WT $p = 7.89e{-}06$, APH 0.4 μM WT vs. APH 0.4 μM *POLQ* −/− $p = 0.5763$, APH 40 μM WT vs. APH 40 μM *POLQ* −/− $p = 0.5761$. **g** Vehicle WT vs. APH 0.4 μM WT $p = 0.0017$, vehicle WT vs. APH 40 uM WT $p = 6.3e{-}05$, APH 0.4 μM WT vs. APH 0.4 μM *POLQ* −/− $p = 0.0041$, APH 40 μM WT vs. APH 40 μM *POLQ* −/− $p = 0.42$. Indel signatures are: Insertions (red) ≥5-bp deletion with flanking microhomologies (MHs) of at least 2 bp (blue), ≥5-bp deletion with flanking MHs of zero or 1 bp (green), short deletion (<5-bp) (purple) and the recurrent deletions in *ASXL1* (**b**, **c**, **d**) *SRSF2* (**e**, **f**, **g**) genes (orange). Source data are provided as a Source Data file.

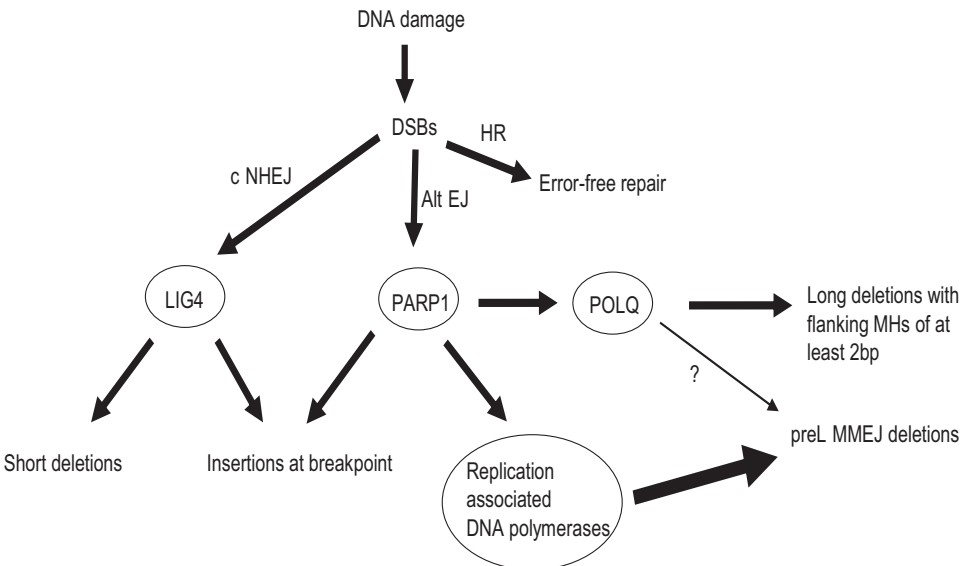

**Fig. 8 preL-MMEJ deletions are the result of alternative DSB repair pathway mediated by PARP1 and replication associated DNA polymerases.** Three distinct pathways of double strand breaks (DSBs) and their consequent repair outcomes. Classical non-homologous end-joining (c-NHEJ) which involves a final ligation step by LIG4, results in short deletions (<5-bp) or insertions at the breakpoint. Homologous recombination (HR) is an error free repair. Both c-NHEJ and alternative end-joining (alt-EJ) pathways are regulated by PARP1. PARP1 is involved in the formation of c-NHEJ related insertions together with the activation of POLQ and other replication associated DNA polymerases. POLQ mediated pathway generates ≥5-bp deletions that are flanked by microhomologies (MHs) of at least 2 bp. However, preL-MMEJ deletions are only mildly contributed by POLQ (thin arrow) while mainly contributed by other replication associated DNA polymerases (thick arrow).

and spun down at 350xg for 10 min. Between 2*10^5 – 1*10^6 cells per reaction were re-suspended in 20 ul SF solution (K562, MARIMO, MOLM-14 and OCI-AML2 cell lines), SE solution (OCI-AML3) or P3 solution (primary CD34 + cells) and added to the RNP complex or 2ug px330 plasmid. FF-120, DN-100, DP-115, DN-100, EO-100 and DZ-100 electroporation programs were used for K562, MARIMO, MOLM-14, OCI-AML2, OCI-AML3 and primary CD34 + cells respectively. Immediately after electroporation, pre-warmed media were added and cells were cultured at the same conditions as the pre-electroporation culturing for additional 2 days (primary CD34 + ) or 4 days (all other cell lines) before they were lysed for NGS sequencing.

**Inhibitors treatment.** In experiments involving MMEJ inhibition, K562 cells were sub-cultured 48 or 2 h before electroporation in a medium containing different dosages of rucaparib camsylate (Sigma, PZ0036) or aphidicolin (Sigma, A4487) respectively. Control cells were sub-cultured in a medium containing vehicle (DMSO). 48 h following electroporation cells were washed with PBS and re-suspended in fresh clean media. Cells were lysed 5 days after electroporation for subsequent NGS sequencing.

**K562 knockout cell lines generation.** K562 cells were electroporated using sgRNA guide targeting *LIG4* or *POLQ* genes followed by a sorting for live single cells using BD FACSMelody™ Cell Sorter (BD Biosciences). Sorted cells were plated onto 96-well plates containing 100ul/well RPMI 1640 Medium with L-Glutamine (Biological Industries, 01-100-1 A), 10% FBS, streptomycin (20 mg/mL) and penicillin (20 unit/mL). 7 days after sorting, 100 ul fresh media were added to each well. Cells were further maintained by replacing 100ul medium from each well once a week. Cell colonies were lysed 28 days after sorting for subsequent NGS sequencing. For cell lysis, cells were spun at 2000g for 10 min, cells pellets were mixed with 30 ul of 50 mM NaOH and heated at 99 °C for 10 min. Then, the reactions were cooled down at room temperature and 2 ul 1 M Tris PH = 8 was added to each reaction. NGS sequencing and analysis were performed as described under sections 7 and 8. Colonies containing bi-allelic frameshift indels at the genomic loci of interest were further isolated and expanded (Supplementary Fig. 12).

**NGS library and targeted sequencing.** For all NGS libraries, we used cell lysis products that served as a template for PCR amplification and library preparations.

Dual indexed illumina Libraries were generated using two-step PCR procedure. 1st PCR primer prefix sequences and 2nd PCR primer sequences were used, similar to a previously described method[46]. All relevant details are as follows: Target-specific primers were designed by Primer3plus (http://www.bioinformatics.nl/cgi-bin/primer3plus/primer3plus.cgi) and were ordered with the described 5' prefixes[46] (IDT). 1st PCR was applied to target the regions of interest. The reaction mixture was composed of a PCR ready mix (using NEBNext® Ultra™ II Q5® Master Mix, NEB, M0544L), a cell lysis product and a final primer concentration of 1uM each. PCR protocol was as follows: 98 °C for 30 s, followed by 40 amplification cycles of 98 °C for 10 s,65 °C for 30 s and a final elongation at 65 °C for 5 min. Following dilution of the 1st PCR products with nuclease free water (1:100), a 2nd PCR was performed using primers composed of Illumina sequencing primers, indexes and adapters, under the same conditions as the 1st PCR with the exceptions of final primer concentration of 0.5uM each and 20 cycles of amplifications. Full sgRNA and primer sequences that were used throughout this study are provided in Supplementary Table 5. Barcoded 2nd PCR products were pooled together at equal volume. Pooled library sizes were selected (2% gel, BluePippin, Sage Science) and sent for 2 × 150-bp deep sequencing (Miseq System, Illumina).

**Variant calling**. 2 × 150-bp pair-end reads deep sequencing data (~5000X depth) from Illumina platform were converted to fastq format. Minimap2.1 algorithm[47] was applied for alignment of the processed fastq files to hg19 genome based targeted sequences resulting in sam files that were further sorted and indexed using pysam 0.15.1 (https://github.com/pysam-developers/pysam). All reads from sorted bam files were assigned to new read groups using picard 2.8.3 'AddOrReplaceReadGroups' command (http://broadinstitute.github.io/picard). In order to avoid misalignments, local realignment was preformed using GATK3.7 'RealignerTargetCreator' and 'IndelRealigner' commands[48]. Mpileup files were generated by samtools 1.8 followed by SNVs and small indels detection using varscan2.3.9 'pileup2cns' command to generate VCF files containing consensus variant calls[49].

**HSPCs cell sorting**. Mononuclear cells (10^6 cells per 100 ul) from peripheral blood samples of two myelofibrosis (MF) patients underwent CD34 enrichment by magnetic beads (Miltneyi Inc.). Both CD34 positive and negative cell fractions underwent fluorescence-activated cell sorting as was previously described[50]. Cells were stained with the following antibodies (all from BD Biosciences unless stated otherwise, catalog numbers and dilution used in parentheses): anti-CD45RA-FITC (555488, 1:25), anti-CD38-PE-Cy7 (335790, 1:200), anti-CD10-Alexa-700 (624040, 1:10), anti-CD7-Pacific Blue (642916, 1:50), anti-CD45-V500 (560777, 1:200), anti-CD34-APC-Cy7 (custom made by BD, CD34 clone 581, 1:100), anti-CD34-PerCP-Efluor 710 (e-Bioscience 46-0344-42, 1:100), anti-CD33-PC5 (Beckman Coulter PNIM2647U, 1:100), anti-CD19-PE (340364, 1:200), anti-CD3-FITC (349201, 1:100), anti-CD56-Alexafluor 647 (557711, 1:100), Streptavidin-QD605 (Invitrogen Q10101MP, 1:200), anti-CD8-APC-H7 (560179, 1:200), anti-light-chain lambda-V450 (561379, 1:200), anti-light-chain kappa-V450 (561327, 1:200), and anti-CD57-APC (555518, 1:200). Subsequently, cells were sorted on a FACSAria III (BD Biosciences) to a post-sort purity of >95%. CD34 enriched cell fraction was gated on CD45 + /CD33- and sorted into the following HSPCs subpopulations: HSC/MPP (CD38-/CD34 + /CD45RA-); MLP (CD38-/CD34 + /CD45RA + ); CMP/MEP (CD38 + /CD34 + /CD7-/CD10-/CD45RA-); and GMP (CD38 + /CD34 + /CD7-/CD10-/CD45RA + ) subsets. CD34 negative cell fraction was sorted into the following mature cell populations: Myeloid cells (CD45dim/CD33 + ); T cells (CD45high/CD3 + /CD8 + ); B cells (CD45high/CD19 + / light chains lambda or kappa + ); and NK cells (CD45high/CD56 + /CD57 + ). DNA from each sorted subpopulation was isolated and amplified using the RepliG whole genome amplification (WGA) kit (REPLI-g Mini Kit for 16 h).

**Xenotransplantation assays**. Animal experiments were performed in accordance to the IACUC of the Weizmann Institute, its relevant guidelines and regulations (11790319-2) and we complied with all relevant ethical regulations for animal testing and research. Eight- to 12-week-old female NOD/SCID/IL-2Rgc-null (NSG) mice were maintained under a 12 h dark/light cycle, at an ambient temperature of around 22 degrees and humidity of 50%. Mice were sub-lethally irradiated (225 cGy) 24 h before transplantation. CD34 + cells were enriched from peripheral blood mononuclear cells of a myelofibrosis (MF) patient by magnetic beads (Miltneyi Inc.) and 50,000 cells were injected into the right femur. Mice were euthanized 16 weeks following transplantation and human engraftment in the injected right femur and non-injected bone marrow (left femur, tibias) was evaluated by flow cytometry analysis using the BD LSR II flow cytometer (BD Biosciences). The threshold for detection of engraftment was 0.1% human CD45 + cells. Human myeloid (human CD45 + /CD33 + /CD19-) and B cells (human CD45 + /CD33-/CD19 + ) were sorted out of the xenografts using the following antibodies (all from BD Biosciences unless stated otherwise, catalog numbers and dilution used in parentheses): anti-CD45-APC (340943, 1:200), anti-CD19-PE (340364, 1:200) and anti-CD33-PE-Cy5 (Beckman Coulter catalog number IM2647U, 1:200). DNA from each sorted subpopulation was isolated and amplified using the RepliG whole genome amplification (WGA) kit (REPLI-g Mini Kit for 16 h).

**ddPCR analysis of graft subpopulations**. ddPCR reaction was performed by using probes designed for *CALR* deletion as described elsewhere[51]. Amplified DNA (2ul from a 1:20 dilution of a 16 h REPLI-g Mini Kit whole-genome amplification, Qiagen) from each sorted population was tested in a 96-well plate in duplicate according to the manufacturer's protocol. Mutant and wild-type sequences were read using a droplet reader with a two-color fluorescein/HEX fluorescence detector (Bio-Rad). The mutant allele frequency was calculated as the fraction of mutant-positive droplets divided by total droplets containing a target. As previously reported[1] the minimum detection level was 1:1,000 (0.1%). Variants were considered present if there were at least three dots in the mutant fluorescein channel resulting in VAF > 0.1%.

**T-cells isolation and expansion from primary AML samples**. CD3 + cells were isolated from peripheral blood mononuclear cells of AML patients by using EasySep Human CD3 Positive Selection Kit II (StemCell Technologies, 17851) and re-suspended in RPMI 1640 Medium with L-Glutamine (Biological Industries, 01-100-1 A), 10% FBS, 250 IU/ml human IL-2 (ThermoFisher scientific, BMS334) and 5 ug/ml anti-CD28 antibody (clone CD28.2, ThermoFisher scientific, 16-0289-81). Re-suspended cells were added to 24-well plate that was pre-coated for 2 h with PBS containing 5 ug/ml anti-CD3 antibody (clone OKT3, ThermoFisher scientific, 16-0037-81). Cells were cultured for 4 days before re-suspension in a fresh RPMI 1640 Medium, 10% FBS, 250 IU/ml hIL-2 and re-plating in a 6-well plate. Cells were then cultured for additional 21 days. Cells purity was assessed by flow cytometry before they were lysed for subsequent NGS sequencing.

**Single cell RNA-seq analyses**. HSPCs RNA-seq profiles were isolated from the HCA immune census BM data based on CD34 expression. A total of 19757 profiles were isolated from the roughly 310,000 BM profiles from 8 different donors. To generate metacells from the profiles, we used the MetaCell package[23] with parameters as specified below. Feature genes were selected using the parameter T_vm = 0.08 and minimal total UMIs of 100, while excluding genes correlated to lateral effects such as mitochondrial genes, immunoglobulin genes, high abundance, prefix "RP-" genes, cell cycle, type I Interferon response and stress. The final feature genes, consisting of 527 genes, were used for the computation of the Metacell balanced similarity graph, with parameters k = 60, n_resamp = 500 and min_mc_size = 20. Outliers threshold of T_lfc = 3.5 was used, with 464 profiles deemed as outliers. Next, we annotated the metacell model using hierarchical clustering of the metacell confusion matrix, supervised analysis of enriched genes and analysis of marker genes (Supplementary Fig. 8). The metacells and profiles were projected and plotted in 2D using mc2d_K = 40, mc2d_T_edge = 0.02 with a max degree of 6, and colored using thresholds on metacells log enrichment scores (lfp values) for marker genes chosen from common markers and the above annotation process.

For studying DNA polymerases genes expression correlations to *PARPs* in HSPCs, we calculated the Pearson correlations using metacells e_gc values calculated once using only HSPCs metacells, and once for all progenitors metacells (Fig. 7a).

**Data analysis**

*Analyses of CRISPR/Cas9 data*. All indels generated by CRISPR/Cas9 system were called using varscan2.3.9. Substitutions and short indels identified in both edited and control samples in *ASXL1*, *SRSF2* and *CALR* loci were excluded. Allele percent was calculated as the number of modified reads associated with each variant divided by the sum of all modified reads per experiment. Overall CRISPR/Cas9 modification frequencies were calculated as the sum of modified reads divided by the mean depth per experiment. We discriminated between three deletion signatures throughout all CRISPR/Cas9 experiments: ≥5 bp deletions with flanking MHs≥2 bp, ≥5 bp deletions with flanking MHs<2 bp and short deletions of <5 bp.

*Analyses of publicly available datasets*. Somatic mutation cohorts were downloaded from publicly available web-links as described under the "Data Availability" section. Only deletions were included in our analyses, duplicate samples were removed from all cohorts. For COSMIC dataset only deletions with available genomic coordinates were analyzed. Additionally, for COSMIC, myeloid deletions were obtained by filtering the 'Primary site' to include 'haematopoietic and lymphoid' tissue following by the exclusion of the letters 'lymph' from the 'Primary histology' column. Identical deletions with multiple 'Mutation CDS' values due to multiple isoforms were combined under a uniform name (for example *ASXL1* c.1888_1910del23 and c.1900_1922del23 were combined under the name *ASXL1* c.1900_1922del23). Deletions from COSMIC dataset that contained common SNPs at adjacent genomic loci, were located at intronic or intergenic regions and those that were reported in a single publication were excluded.

To exclude common SNPs from all sequencing cohorts, minor allele frequencies (MAF) for each deletion were identified using Annovar tool (https://github.com/WGLab/doc-ANNOVAR) according to the following datasets: AF, ExAC_ALL, Kaviar_AF, ExAC_nonpsych_ALL and AF_popmax. Variants with MAF of 0.0001 or above in at least one dataset were filtered out. Deletions with no available data in any of these datasets were included. For signature detection, deletion coordinates with flanking 20 bp from both deletion ends (e.g start-20bp, end+20 bp) were

generated and used as an input for bedtools getfasta command to generate Fasta files for all deletions flanking sequences. 'MH signature' detection was performed using an in-house matlab code by analyzing each deletion's flanking sequences for microhomologies (MHs). Specifically, we discriminated between three deletion signatures throughout all data analyses: ≥5 bp deletions with flanking MHs≥2 bp, ≥5 bp deletions with flanking MHs<2 bp and short deletions of <5 bp. Of note, our detection algorithm did not discriminate between short deletion that are located in microsatellite repeats and those that are not, as this was beyond the scope of the current manuscript. Matlab code is available as described under the "Code Availability" section.

All other analyses were performed using R (version 3.5.2).

**Reporting summary**. Further information on research design is available in the Nature Research Reporting Summary linked to this article.

## Data availability

Raw Illumina sequencing reads associated with CRISPR/Cas9 cell line experiments have been deposited in the NCBI Short Read Archive under bioproject ID PRJNA707245. Publicly available datasets used in this study are available in the following web links: https://cancer.sanger.ac.uk/cosmic/download (COSMIC dataset), https://www.nejm.org/doi/full/10.1056/nejmoa1516192 (1540 adult-AML dataset[10]), https://www.nejm.org/doi/full/10.1056/NEJMoa1716614 (2045 MPN dataset[11]), http://www.vizome.org/aml/ (BeatAML dataset[12]). For 1540 adult-AML and 2045 MPN datasets, we used annotated mutational data that are open-access and available for download as part of the supplementary appendixes of these papers (table s5 in AML paper[10] and table s4 in MPN paper[11]). Full data containing the deletion signatures from the publicly available datasets as well as CRISPR/Cas9 indel data are provided as a Source Data file. All relevant data are also available from the corresponding author upon reasonable request. Source data are provided with this paper.

## Code availability

MMEJ deletion matlab code is documented on GitHub (https://github.com/ShlushLab/MMEJ_detection) and is publicly available under the MIT license from the following Zenodo repository (https://doi.org/10.5281/zenodo.4555395)[52]. For the Metacell analysis, we used the previously published MetaCell package[23] with parameters specified under the "Methods" section. Metacell code is documented on GitHub (https://github.com/tanaylab/metacell) and is publicly available under the MIT license from the following Zenodo repository (https://doi.org/10.5281/zenodo.3334525)[53].

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

## Acknowledgements
The authors wish to thank Prof. John Dick, and Dr. Ayal Hendel for fruitful discussion and support. All primary patient samples that were used in this study were generously provided by Dr. Mark Minden and through the Leukemia Tissue Bank at Princess Margaret Cancer Centre/ University Health Network. L.S. is the incumbent of The Ruth and Louis Leland career development chair. This research was supported by the EU horizon 2020 grant project MAMLE ID: 714731, LLS and rising tide foundation Grant ID: RTF6005-19, ISF-NSFC 2427/18, ISF-IPMP-Israel Precision Medicine Program 3165/19, BIRAX 713023, the Ernest and Bonnie Beutler Research Program of Excellence in Genomic Medicine, awarded to LIS. LIS is an incumbent of the Ruth and Louis Leland career development chair. N.K. is an incumbent of the Applebaum Foundation Research Fellow Chair. This research was also supported by the Sagol Institute for Longevity Research, the Barry and Eleanore Reznik Family Cancer Research Fund, Steven B. Rubenstein Research Fund for Leukemia and Other Blood Disorders, the Rising Tide Foundation and the Applebaum Foundation.

## Author contributions
T.F. designed and developed the study, performed CRISPR/Cas9 experiments, cells culture and maintenance, deep targeted sequencing, analyzed sequencing data, performed variant calling, analyzed the publicly available datasets and wrote the manuscript. A.B. performed single-cell RNA analysis. Y.M. performed CRISPR/Cas9 experiments, cells culture and maintenance, deep targeted sequencing. N.I.C. provided bioinformatics support and wrote the matlab code for MMEJ detection. N.K. revised the paper and contributed to data interpretation. T.B. provided sequencing and technical support. A.M. and J.M. Performed xenotransplantation experiments, cell sorting and ddPCR. M.D.M. and G.V. enabled sample acquisition, M.M. and Z.L. helped with the knockout experiments, A.T. supervised single-cell RNA analysis L.I.S. designed and supervised the study and wrote the manuscript.

## Competing interests
The authors declare no competing interests.
