## [Peer Review File · Nature Communications]

REVIEWER COMMENTS

Reviewer #1 (Remarks to the Author): expert in genomics of leukaemia and clonal hematopoiesis

In the manuscript by Feldman et al, entitled "Recurrent deletions in clonal hematopoiesis are the result of microhomology-mediated end joining of DNA double strand breaks", the authors identify regions of microhomology surrounding recurrent deletions in SRSF2, ASXL1, and CALR in myeloid malignancies. Deletions in these genes could be generated by creating DSBs near these sites in cell lines and primary samples. Manipulation of PARP1 or LIG4 could modulate the generation of the deletions, implicating the MMEJ pathway in their generation.

Comments.

1. Deletions were identified in T cells. What was the purity of the sorted samples to ensure the signal is not coming from contaminating myeloid cells?
2. These mutations in ASXL1 and SRSF2 become more common as people age, yet the induction of the mutations following DSBs using CRISPR was similar in young vs. aged CD34+ cells. Do aged HSCs have more dysregulated MMEJ or HR pathways compared to young HSCs?
3. The presentation of the single cell data is confusing. The authors state the "RNA-seq data of human HSCs suggest that the MMEJ pathway is activated as HSCs exit quiescence". These results are underdeveloped and not convincing. What do the yellow dots represent in SF6?

Reviewer #2 (Remarks to the Author): expert in DNA repair, MH, NHEJ

Here Feldman et al. use COSMIC, BeatAML, and TCGA data, as well as functional follow-up analyses to investigate the relationship between recurrent deletions in clonal hematopoiesis and microhomology-mediated end joining (MMEJ) of DNA double strand breaks. They initially identify CALR, ASXL1 and SRSF2, common deletions in myeloid malignancies sharing and MH-based signature using COSMIC. In order to replicate their findings, they extend their analysis to primary samples from two additional cohorts, and they find additional evidence of frequency of these deletions. They next move to experimentally validate these bioinformatic findings by introducing DSBs using the CRISPR Cas9 system around the hotspot regions of the CALR, ASXL1 and SRSF2 genes in K562 CML cell line. They were able to recapitulate certain ASXL1 and SRSF2 genes MH-based deletions, but not CALR in the genes of interest.

K562 is known to have the BCR-ABL oncoprotein that drives this disease. Are the MH-dependent deletions on the presence of BCR-ABL? Additionally, this cell line has basal increases in MMEJ, according to Sallmyr et al. Therefore, are the authors biasing for this MMEJ deletion outcome? Recapitulation of these deletions in actual denovo AML cell lines representing other myeloid malignancies, and without increased basal ALT NHEJ/MMEJ is required to strengthen the conclusions the authors are making.

Reviewers show that MMEJ deletions decrease when PARP inhibitors are used. Can the reviewers show that MMEJ is functionally decreased by performing functional MMEJ activity assays, and that consequent MMEJ deletions emerge? PARP1 is involved in ss break repair as well as MMEJ. Authors may consider knocking down/out polq and measuring functional MMEJ and inducing specific deletions.

MMEJ is a backup pathway that comes into play when other DSB repair pathways fail. While C-NHEJ has been tested, are these deletions more frequent in backgrounds of HR deficiency? Therefore,

deletion of BRCA or induction of BRCAness in these cells is also required.

The authors go on to compare AML to solid tumors and find that there are similar signatures between AML and some HR deficient solid tumors. Again, the authors need to show experimentally that MMEJ deletions occur in solid tumors by mechanistically investigating these tumors.

To study this phenomenon and its relation to MMEJ, the authors first interrogate sequencing data from healthy individuals and pre-AML cases. They then use a mouse model to show that multipotent HSPCs upregulate MMEJ associated genes. All of the above questions apply to this work.

While the authors findings of microhomology-mediated end joining (MMEJ) of DNA double strand breaks in ASXL1 and SRSF2 genes are interesting they have not sufficiently mechanistically proved how this occurs.

Reviewer #3 (Remarks to the Author): expert in hematopoiesis genomics and single-cell RNA-seq

In their manuscript entitled “Recurrent deletions in clonal hematopoiesis are the result of Microhomology-mediated end joining of DNA double strand breaks” Feldman et al., provide a set of in vitro data and genomic analyses that implicate MH-mediated end joining in the mutational patterns seen in clonal hematopoiesis and leukemia development. The initial hypothesis and description of datasets are very interesting and there appears to be a clear pattern of interest in several known driver genes (CALR, ASXL1, SRSF2). That said, I found the manuscript could have gone further to provide more evidence that this is a common mechanism in clonal hematopoiesis or leukemogenesis. I have tried to highlight my concerns below:

1) Over-reliance on a single cell line – The large majority of mechanistic work is performed in K562 cells and show that CRISPR-induced DSBs can be repaired in an MH-mediated manner for 2 of 3 driver genes assessed. CALR mutations are the most common deletion with an MH-based signature by far (Fig 1) and DSBs do not result in MH-mediated repair, challenging the model cell line system. This is especially important since the abstract claims “We demonstrate that these MH-based deletions are the result of double strand breaks (DSBs) followed by Microhomology-mediated end joining (MMEJ) repair” with little to no evidence at all that this is the mechanism in CALR-mutated patients (aside from a tiny fraction in the Lig4 KO).

It would be nice to see these findings replicated in multiple cell lines or (preferably) in primary human samples. The authors do perform one experiment (Ext Fig 4) in human samples for ASXL1, but they do not report the overall frequency of the indel event (just the frequency of total indels), making it really difficult to know how common this is and whether or not it would apply to other mutations remains unclear. It would be much stronger to see this performed in more primary samples with different mutations and to include the overall DSB to indel rates in these experiments. As it stands, it is an interesting proof-of-concept experiment (which is a strength), but I think it is a bit preliminary to make the conclusion so strongly.

2) Low numbers of patient samples – the authors use 2 CALRdel52 MF patients, one of which went into a xenograft for Figure 6A. This is far too few patients to be able to conclude anything in a robust manner, especially considering there are numerous MPN labs who likely already have the relevant information

3) Analysis that builds a case rather than tests a hypothesis – Figures 1, 3, 5, and 6B/C are all derived

from previously published datasets. While I am certainly not against using publicly available datasets to bolster the conclusions of a paper, I am concerned that this is done in a somewhat selective manner. For example, the gene expression data presented in Figure 6B/C shows that MMEJ related genes are activated upon entry of HSPCs into cell cycle, but dozens of other gene sets would also be upregulated upon cell cycle entry and dozens of other cell types would likely upregulate MMEJ-related genes in quiescence versus cell cycle. Without some grounding in these other comparisons, it is impossible to assess the likelihood that this is not a spurious third variable.

Also, the comparison in Figure 5A could be substantially improved by first reporting deletion frequency out of the total rather than making these comparisons within deletion types – the current comparison is just as easily skewed by an enrichment in short deletions as it is by an increase in long-deletions within a particular cancer type. The current display masks all of this data which is critical for understanding how real this phenomenon is. Also, why are no comparisons made to other non-HR deficient tumors (which are also not significantly different to the BeatAML distribution)? It seems really unfair to say that AML is similar to one side of the graph and not the other when the statistical tests suggest that most deletion ratios are not very different across most tumor types. Either way, the deletion ratio metric is potentially flawed if absolute frequencies of deletions across tumor types are not the same (which I would imagine they are not).

4) The authors say “While we cannot rule out a selective advantage for the MH-based deletion specifically in the hematopoietic system, the most likely interpretation of our results is that specific mutational mechanisms contribute to the generation of the MH-based deletion in myeloid malignancies” – I do not see any evidence at all supporting this statement and would rather suggest that a selective advantage in the blood system could very possibly explain this result. If the authors want to support their statement, it would be better to see the same DSB CRISPR experiment in another primary tissue type and show that the MH-mediated repair does NOT occur.

Minor points:

Title – based on the current version and data in this manuscript, I think it is unfair to state that clonal hematopoiesis mutations ARE the result of MH-mediated end joining. Perhaps this could be softened to “MH-mediated end joining as a potential driver of clonal hematopoiesis mutations”

Comparison to Cosmic data – I wasn't 100% clear on what was being compared in Figure 3. If I understand it correctly, there were 1434 ASXL1 mutant hematological tumors of which 376 had deletions and 153 of these were the long deletion (orange bar) compared to 252 ASXL1 mutant non-hematological tumors, of which 103 had deletions and 4 of these were the long deletion (orange bar). If this is correct, it would be much more helpful to explain this in the figure legend or text together rather than having to go back/forth to work out. It would also be interesting to know whether the 4 cases in the latter group were from the same cancer subtype or randomly spread? It is also not really appropriate to use a chi-squared test here to compare amongst deletion types as described above (lines 86-87 – the real comparison should be the overall frequency per case as increases in another deletion type could equally skew the frequency (and likely do by the looks of Fig 3b and the large purple bar).

Response Figure 2 was taken from Biechonski, S., Olender, L., Zipin-Roitman, A. et al. Attenuated DNA damage responses and increased apoptosis characterize human hematopoietic stem cells exposed to irradiation. *Sci Rep* 8, 6071 (2018). <https://doi.org/10.1038/s41598-018-24440-w> under the following license: <https://creativecommons.org/licenses/by/4.0/>

Response Figure 5 was taken from Bennardo, N., Cheng, A., Huang, N., and Stark, J.M. Alternative-NHEJ Is a Mechanistically Distinct Pathway of Mammalian Chromosome Break Repair. *PLoS Genet* 4(6): e1000110 (2008). <https://doi.org/10.1371/journal.pgen.1000110> under the following license: <https://creativecommons.org/licenses/by/4.0/>

Jan-06-2021

Response to reviewers comments:

Reviewer #1 (Remarks to the Author): expert in genomics of leukaemia and clonal hematopoiesis

In the manuscript by Feldman et al, entitled “Recurrent deletions in clonal hematopoiesis are the result of microhomology-mediated end joining of DNA double strand breaks”, the authors identify regions of microhomology surrounding recurrent deletions in SRSF2, ASXL1, and CALR in myeloid malignancies. Deletions in these genes could be generated by creating DSBs near these sites in cell lines and primary samples. Manipulation of PARP1 or LIG4 could modulate the generation of the deletions, implicating the MMEJ pathway in their generation.

Comments.

1. Deletions were identified in T cells. What was the purity of the sorted samples to ensure the signal is not coming from contaminating myeloid cells?

Response

We thank the reviewer for this comment. As described in the revised version under the methods section, all T-cells that were used in this study were separated from AML samples by CD3 positive selection kit, followed by CD3/CD28 stimulation. T-cells were then cultured for 3 weeks in vitro and assessed by flow cytometry prior to cell lysis and sequencing. Representative flow cytometry analysis is shown below, indicating 98-100% purity of CD45 positive CD3 positive CD33 negative population.

Response Figure 1.

2. These mutations in *ASXL1* and *SRSF2* become more common as people age, yet the induction of the mutations following DSBs using CRISPR was similar in young vs. aged CD34+ cells. Do aged HSCs have more dysregulated MMEJ or HR pathways compared to young HSCs?

Response

We thank the reviewer for this important comment. Our results do not provide direct evidence that aged HSCs have dysregulated repair pathways and as the reviewer correctly mentioned, all primary samples and cell lines that were used in the revised version (revised Fig. 4 c, d, e, f) showed similar repair outcomes following DSBs. We have added a discussion point in the revised manuscript regarding this comment and suggest a possible experimental route of how to resolve it.

pp. 13, line 276

“PreL-MMEJ deletions are typically identified among the elderly. An important factor contributing to DSB repair choice, might be the age of the cell of origin in which the DSBs occur. In our CRISPR/Cas9 based model, similar frequencies of preL-MMEJ deletions were obtained in young and aged human HSPCs. This might be due to the fact that our model system is not mimicking the exact biological context in which preL-MMEJ arise. It remains unclear whether preL-MMEJ deletions can occur in HSCs at any age and expand due to selective advantage at older age or that preL-MMEJ deletions preferentially occur in aged HSCs. To elucidate this, the phylogenetic origins of preL-

MMEJ deletions can be studied in single cells as was previously done³⁵ to determine the exact age in which they originate “

We think that the first step should be to understand whether the deletions can occur at young age and provide selective advantage at old age, or as the reviewer suggest that they occur at old age to begin with.

3. The presentation of the single cell data is confusing. The authors state the “RNA-seq data of human HSCs suggest that the MMEJ pathway is activated as HSCs exit quiescence”. These results are underdeveloped and not convincing. What do the yellow dots represent in SF6?

Response

We thank the reviewer for this important comment. We fully agree that the presentation of the RNA-seq data by itself is confusing and not convincing. Therefore, we decided to use these figures as extended data figures (revised Extended Data Fig. 9, 10c). In the revised version, we used the single cell data to raise specific hypotheses directly relevant to our experimental results. This suggested a correlation between *PARP1* and replicative polymerases in HSCs (revised Fig. 7a). We then provide experimental evidence that the MMEJ is dependent on cell replication and is inhibited by aphidicolin treatment (revised Fig. 7). These experiments were performed due to the data obtained from the single-cell RNA-seq.

Reviewer #2 (Remarks to the Author): expert in DNA repair, MH, NHEJ

1. Here Feldman et al. use COSMIC, BeatAML, and TCGA data, as well as functional follow-up analyses to investigate the relationship between recurrent deletions in clonal hematopoiesis and microhomology-mediated end joining (MMEJ) of DNA double strand breaks. They initially identify CALR, ASXL1 and SRSF2, common deletions in myeloid malignancies sharing and MH-based signature using COSMIC. In order to replicate their findings, they extend their analysis to primary samples from two additional cohorts, and they find additional evidence of frequency of these deletions. They next move to experimentally validate these bioinformatic findings by introducing DSBs using the CRISPR Cas9 system around the hotspot regions of the CALR, ASXL1 and SRSF2 genes in K562 CML cell line. They were able to recapitulate certain ASXL1 and SRSF2 genes MH-based deletions, but not CALR in the genes of interest. K562 is known to have the BCR-ABL oncoprotein that drives this disease. Are the MH-

dependent deletions on the presence of BCR-ABL? Additionally, this cell line has basal increases in MMEJ, according to Sallmyr et al. Therefore, are the authors biasing for this MMEJ deletion outcome? Recapitulation of these deletions in actual denovo AML cell lines representing other myeloid malignancies, and without increased basal ALT NHEJ/MMEJ is required to strengthen the conclusions the authors are making.

Response

We thank the reviewer for this very important comment. As the reviewer correctly mentioned, K562 cells were shown to upregulate MMEJ (Sallmyr et al) while downregulating HR (Podszylalow-Bartnicka et al 2014). Following the reviewer's suggestion, we repeated the experiments in four additional AML cell lines (revised Fig. 4c, d). Additionally, as we provide evidence that preL-MMEJ deletions originate in multipotent HSCs, we repeated our experiments in isolated CD34+ cells derived from six individuals (revised Fig. 6e, f).

2. Reviewers show that MMEJ deletions decrease when PARP inhibitors are used. Can the reviewers show that MMEJ is functionally decreased by performing functional MMEJ activity assays, and that consequent MMEJ deletions emerge?

Response

We thank the reviewer for this comment. In order to perform functional MMEJ activity we used EJ2-GFP based reporter (Bennardo N et al., PLOS Genetics 2008). This reporter system is relying on I-SceI mediated DSBs. In this system, only cells in which DSBs are repaired by MMEJ and annealing of specific microhomology fragments is taking place, would release GFP. Please see the illustration below (taken from S Biechonski et al 2018):

Response Figure 2.

We used this reporter assay with and without I-SceI. Additionally, positive control GFP expression plasmid (CMV-GFP) was used to exclude the possibility of technical limitations. This experiment demonstrated that 48h after transfection, only 0.5 % of the living cells released GFP, while 61% released GFP in the positive control (Response Fig. 3).

Response Figure 3.

Similar results were obtained 72h post-transfection:

Response Figure 4.

These results are in line with the original paper describing this E2J reporter system (Bennardo N et al., PLOS Genetics 2008). Please see the Response Figure 5 taken from Bennardo N et al 2018, describing efficiency of around 0.5% among WT cells:

Response Figure 5.

According to our results and others, CRISPR/Cas9 mediated DSBs demonstrate MMEJ repair efficiency that sometimes accounts for more than 50% of total modified alleles (revised Fig. 5b). This is in part due to the fact that CRISPR/Cas9 mediated DSBs occur in endogenous genomic loci and that the readout is not biased to annealing of specific microhomology fragments. We therefore concluded that our model system may be a more sensitive tool to study the mechanism behind preL-MMEJ deletions. The low levels of MMEJ in the reported system makes it harder to study.

3. PARP1 is involved in ss break repair as well as MMEJ. Authors may consider knocking down/out polq and measuring functional MMEJ and inducing specific deletions.

Response

We thank the reviewer for this important comment. We fully agree that PARP1 plays a regulatory role in MMEJ and is not specific to MMEJ. As we demonstrate (revised Fig. 5), high concentration of PARP1 inhibitor result in a decreased frequency of c-NHEJ mediated insertions, suggesting an involvement of PARP1 also in c-NHEJ. Following the reviewer's comment, we generated three distinct POLQ $-/-$ cells as described in the revised results.

pp. 9, Line 187

"We generated three distinct K562 POLQ $-/-$ cells harboring frameshift mutations at exon 14, 16 and 18 of the POLQ gene (Extended Data Figure 7). Each one of these mutations presumably leads to a premature stop codon upstream or inside the polymerase domain, previously shown to be required for end joining repair^{21,22}."

We demonstrated an increase in c-NHEJ mediated deletions (revised Fig. 6f) together with a decreased fractions of MH-based deletions (revised Fig. 6g) in POLQ $-/-$ cells, providing functional evidence for the role of polymerase theta in MMEJ. However, these results highlight the potential involvement of other polymerases in generating preL-MMEJ deletions, as the recurrent MMEJ deletions in both ASXL1 and SRSF2 did not significantly change in the POLQ $-/-$ cells (revised Fig. 6h).

4. MMEJ is a backup pathway that comes into play when other DSB repair pathways fail. While C-NHEJ has been tested, are these deletions more frequent in backgrounds of HR deficiency? Therefore, deletion of BRCA or induction of BRCAness in these cells is also required.

Response

We thank the reviewer for this comment. Following this comment we generated K562 cells with the BRCA1 homozygote frameshift mutations c.5263dupT:p.S1755fs. This single base insertions is three nucleotides downstream from the single base insertion p.Q1756fs in BRCA1 gene. HCC1937 cells harboring p.Q1756fs mutations are considered BRCA1 deficient and were shown to be sensitive to PARP inhibitors (Katherine Sullivan-Reed et al, 2018).

In this model, BRCA1 $-/-$ cells demonstrate an enhanced c-NHEJ activity in both ASXL1 and SRSF2 loci without an increase in preL-MMEJ deletions (Response Fig. 6).

Response Figure 6.

SRSF2

ASXL1

These results indicate that in BRCA1 $-/-$ background, there is a relative increase in c-NHEJ and that BRCA1 deficiency alone cannot explain HSCs repair shift towards MMEJ. As we do not provide enough evidence to support HR deficiency as driving preL-MMEJ deletions in HSCs we will not show this data in the manuscript.

5. The authors go on to compare AML to solid tumors and find that there are similar signatures between AML and some HR deficient solid tumors. Again, the authors need to show experimentally that MMEJ deletions occur in solid tumors by mechanistically investigating these tumors.

Response

We thank the reviewer for this comment. Following our BRCA1 KO results and the reviewer's comments, we realized that we do not have enough experimental data to support the role of HR-deficiency as a driver of preL-MMEJ deletions. We therefore decided to exclude the comparison between AML and solid tumors from the current manuscript.

6. To study this phenomenon and its relation to MMEJ, the authors first interrogate sequencing data from healthy individuals and pre-AML cases. They then use a mouse model to show that multipotent HSPCs upregulate MMEJ associated genes. All of the above questions apply to this work.

We do not fully understand this comment.

7. While the authors findings of microhomology-mediated end joining (MMEJ) of DNA double strand breaks in ASXL1 and SRSF2 genes are interesting they have not sufficiently mechanistically proved how this occurs.

Response

We thank the reviewer for this comment. Following the reviewer's concerns we generated POLQ $-/-$ cells and provided evidence that some MMEJ deletions are dependent on polymerase theta, however preL-MMEJ could be generated in a similar efficacy in POLQ $-/-$ cells compared to WT cells.

As we describe POLQ independent MMEJ pathway, we then focused our RNA-seq analysis to gain insights into additional polymerases that may be participating in MMEJ (Fig 7a).

pp. 9 Line 190

“In *SRSF2*, a significant decrease in total fractions of MH-based deletions together with an increase in short deletions, validated a role for polymerase theta in MMEJ (Fig. 6f, g). In contrast, *POLQ* knock out resulted in a mild and mostly insignificant decrease in the fractions of both preL-MMEJ deletions (Fig. 6c, h, Extended Data Fig. 7, Supplementary Table 9). This suggests that polymerase theta has a limited role in the pathway leading to preL-MMEJ deletions. We therefore hypothesized that other DNA polymerases may collaborate with *PARP1* and involved in the pathway leading to preL-MMEJ deletions in humans. In order to identify such an involvement, we analyzed gene expression data of single human HSCs.

Inhibition of replicative DNA polymerases by aphidicolin reduces the formation of preL-MMEJ deletions

We next studied the gene expression profiles of human single bone-marrow (BM) progenitor cells as was previously described²³. We analyzed BM CD34+ profiles from the Human Cell Atlas Consortium’s immune census dataset (<https://preview.data.humancellatlas.org/>) (Extended Data Fig. 8) and focused on multipotent HSCs expressing CD34 and AVP markers, and proliferating MPPs (cells of origin of MH-based deletions) (Extended Data Fig. 9). We noticed that as HSCs enter cell replication, they upregulate components of the c-NHEJ, MMEJ, and HR pathways (Extended Data Fig. 10). Our experimental results demonstrated that inhibition of *PARP1* by rucaparib camsylate resulted in a decreased production of preL-MMEJ deletions *in vitro*. As we also provide evidence that the preL-MMEJ deletions originate in multipotent HSCs, we assessed for a possible correlation between the expression levels of *PARPs* and a list of human DNA polymerases²⁴ specifically in HSCs and MPPs, for polymerases that are not correlated throughout all progenitor states. Among this sub-population, *PARP1* expression levels were shown to significantly correlate only in HSCs with *POLQ*, but also with *POLD1*, *POLE* and *POLE4* gene expression levels (Fig. 7a).”

Lastly, MMEJ inhibition following aphidicolin treatment (revised Fig. 7b-g) exposed the unique correlation between MMEJ and cell replication. This may imply that replicative DNA polymerases are directly active during MMEJ, a model not previously suggested, or that stalling of DNA replication gave advantage to c-NHEJ over MMEJ. We offer these two optional explanations in the revised discussion:

pp. 13 Line 273

“However, future studies are warranted to assess whether aphidicolin related reduction of preL-MMEJ is due to a direct inhibition of replicative polymerases or as a consequence of cell cycle arrest.”

All of the above may shed light on the biological contexts driving early mutagenesis in preL HSCs. A better understanding of such contexts may aid in future treatment and prevention of clonal hematopoiesis.

Reviewer #3 (Remarks to the Author): expert in hematopoiesis genomics and single-cell RNA-seq

In their manuscript entitled “Recurrent deletions in clonal hematopoiesis are the result of Microhomology-mediated end joining of DNA double strand breaks” Feldman et al., provide a set of in vitro data and genomic analyses that implicate MH-mediated end joining in the mutational patterns seen in clonal hematopoiesis and leukemia development. The initial hypothesis and description of datasets are very interesting and there appears to be a clear pattern of interest in several known driver genes (CALR, ASXL1, SRSF2). That said, I found the manuscript could have gone further to provide more evidence that this is a common mechanism in clonal hematopoiesis or leukemogenesis. I have tried to highlight my concerns below:

1) Over-reliance on a single cell line – The large majority of mechanistic work is performed in K562 cells and show that CRISPR-induced DSBs can be repaired in an MH-mediated manner for 2 of 3 driver genes assessed. CALR mutations are the most common deletion with an MH-based signature by far (Fig 1) and DSBs do not result in MH-mediated repair, challenging the model cell line system. This is especially important since the abstract claims “We demonstrate that these MH-based deletions are the result of double strand breaks (DSBs) followed by Microhomology-mediated end joining (MMEJ) repair” with little to no evidence at all that this is the mechanism in CALR-mutated patients (aside from a tiny fraction in the Lig4 KO).

It would be nice to see these findings replicated in multiple cell lines or (preferably) in primary human samples. The authors do perform one experiment (Ext Fig 4) in human samples for ASXL1, but they do not report the overall frequency of the indel event (just the frequency of total indels), making it really difficult to know how common this is and whether or not it would apply to other mutations remains unclear. It would be much stronger to see this performed in more primary samples with different mutations and to include the overall DSB to indel rates in these experiments. As it stands, it is an interesting proof-of-concept experiment (which is a strength), but I think it is a bit preliminary to make the conclusion so strongly.

Response

We thank the reviewer for this comment. We fully agree that over-reliance on a single cell line may be problematic. Following the reviewer’s comment we replicated our experiments in four additional AML cell lines (revised Fig. 4c, d) and CD34+ cells isolated from six primary human samples (Fig. 4e, f). Additionally, as the reviewer’s suggested, we added figures describing the overall indel frequencies per experiment and the

relative contribution of each signature (For example revised Fig. 5b, e).

2) Low numbers of patient samples – the authors use 2 CALRdel52 MF patients, one of which went into a xenograft for Figure 6A. This is far too few patients to be able to conclude anything in a robust manner, especially considering there are numerous MPN labs who likely already have the relevant information

Response

We thank the reviewer for this comment. We agree that two samples are too few patients to be able to draw substantial conclusions. However, as described in the text, previous reports provided evidence that CALR MH-based deletions originate in multipotent HSCs (Nangalia et al, 2013). We wished to provide a validation for this finding.

pp. 6 Line 116

“Somatic mutations in *CALR*, *ASXL1* and *SRSF2* genes have been shown by others to be pre-leukemic lesions originating in early multipotent hematopoietic stem cells^{14 15}. We wished to validate that multipotent HSCs are the cells of origin for the three recurrent MH-based deletions in these genes.”

Regarding the origins of *ASXL1* and *SRSF2* MH-based deletions, we isolated T-cells from five additional AML samples (revised Fig. 3a) and identified the mutations in low VAFs, suggesting multipotent HSCs as the cell of origin.

3) Analysis that builds a case rather than tests a hypothesis – Figures 1, 3, 5, and 6B/C are all derived from previously published datasets. While I am certainly not against using publicly available datasets to bolster the conclusions of a paper, I am concerned that this is done in a somewhat selective manner. For example, the gene expression data presented in Figure 6B/C shows that MMEJ related genes are activated upon entry of HSPCs into cell cycle, but dozens of other gene sets would also be upregulated upon cell cycle entry and dozens of other cell types would likely upregulate MMEJ-related genes in quiescence versus cell cycle. Without some grounding in these other comparisons, it is impossible to assess the likelihood that this is not a spurious third variable.

Response

We thank the reviewer for this comment. We agree that analysis of published datasets should ideally test a hypothesis and that we might have over-used some of these analyses. Following the reviewer’s comment we decided to transfer the RNA-seq heat

map and correlation plot to the extended data (revised Extended Data Fig. 9, 10c). Additionally, based on our experimental data indicating that preL-MMEJ deletions can be generated in POLQ $-/-$ cells (Fig. 6), we used the RNA-seq data to test a specific hypothesis that other polymerases may be correlated with PARP1 expression in replicating HSCs (revised Fig 7a). We then provide experimental evidence that the MMEJ is inhibited by aphidicolin treatment (revised Fig. 7). These experiments were performed due to the data obtained from the single-cell RNA-seq.

Also, the comparison in Figure 5A could be substantially improved by first reporting deletion frequency out of the total rather than making these comparisons within deletion types – the current comparison is just as easily skewed by an enrichment in short deletions as it is by an increase in long-deletions within a particular cancer type. The current display masks all of this data which is critical for understanding how real this phenomenon is. Also, why are no comparisons made to other non-HR deficient tumors (which are also not significantly different to the BeatAML distribution)? It seems really unfair to say that AML is similar to one side of the graph and not the other when the statistical tests suggest that most deletion ratios are not very different across most tumor types. Either way, the deletion ratio metric is potentially flawed if absolute frequencies of deletions across tumor types are not the same (which I would imagine they are not).

Response

We fully agree with this important comment. Following this comments and others, we aimed to experimentally test whether HR-deficiency can over activate the MMEJ pathway. We generated K562 cells with the BRCA1 homozygote frameshift mutations c.5263dupT:p.S1755fs, in close proximity to the single base insertion p.Q1756fs in BRCA1 gene. HCC1937 cells harboring p.Q1756fs mutations are considered BRCA1 deficient and were shown to be sensitive to PARP inhibitors (Katherine Sullivan-Reed et al, 2018).

In this model BRCA1 KO cells demonstrate an enhanced c-NHEJ activity in both ASXL1 and SRSF2 loci. No reduction in the frequencies of preL-MMEJ deletion was observed in BRCA1 $-/-$ cells. (Response Fig. 6).

These results indicate that BRCA1 deficiency alone cannot explain HSCs repair shift towards MMEJ. As we do not provide enough evidence to support HR deficiency as driving preL-MMEJ deletions in HSCs, and as the comparison between AML and solid tumors may be misleading as the reviewer correctly suggested, we decided to exclude this figure from the current manuscript.

4) The authors say “While we cannot rule out a selective advantage for the MH-based deletion specifically in the hematopoietic system, the most likely interpretation of our results is that specific mutational mechanisms contribute to the generation of the MH-based deletion in myeloid malignancies” – I do not see any evidence at all supporting this statement and would rather suggest that a selective advantage in the blood system could very possibly explain this result. If the authors want to support their statement, it would be better to see the same DSB CRISPR experiment in another primary tissue type and show that the MH-mediated repair does NOT occur.

Response

We thank the reviewer for this comment. We induced specific DSBs into ASXL1 locus in BJ fibroblasts and observed similar indel pattern (data not shown). Accordingly, we believe that our CRISPR/Cas9 model system is not accurately mimicking the biological scenario in which DSBs occur in HSCs and generate MMEJ deletions at high efficiency. The fact that preL-MMEJ deletions are unique to myeloid malignancies while other somatic mutations in the same genomic region occur in other tumors might suggest different mutational mechanisms.

As we agree with the reviewer that this is not a definitive proof, we have changed the text accordingly:

pp.5 Line 110

“While we cannot rule out a selective advantage for the MH-based deletion truncated ASXL1 protein specifically in the hematopoietic system, a possible interpretation of these results is that specific mutational mechanisms contribute to leukemogenesis in the myeloid malignancies’ cell of origin.”

Minor points:

Title – based on the current version and data in this manuscript, I think it is unfair to state that clonal hematopoiesis mutations ARE the result of MH-mediated end joining. Perhaps this could be softened to “MH-mediated end joining as a potential driver of clonal hematopoiesis mutations”

We thank the reviewer for this comment. However, in light of the new experimental data we provide in the revised manuscript, we are confident that the revised title: “Recurrent deletions in clonal hematopoiesis are driven by Microhomology-mediated end joining” summarizes the results accurately.

Comparison to Cosmic data – I wasn’t 100% clear on what was being compared in Figure 3. If I understand it correctly, there were 1434 ASXL1 mutant hematological tumors of

which 376 had deletions and 153 of these were the long deletion (orange bar) compared to 252 ASXL1 mutant non-hematological tumors, of which 103 had deletions and 4 of these were the long deletion (orange bar). If this is correct, it would be much more helpful to explain this in the figure legend or text together rather than having to go back/forth to work out. It would also be interesting to know whether the 4 cases in the latter group were from the same cancer subtype or randomly spread? It is also not really appropriate to use a chi-squared test here to compare amongst deletion types as described above (lines 86-87 – the real comparison should be the overall frequency per case as increases in another deletion type could equally skew the frequency (and likely do by the looks of Fig 3b and the large purple bar).

In Figure 3, among 1434 unique hematological samples analyzed, 376 had deletions and 153 of which were the recurrent MH-based deletions. This is compared to 4/103 unique cases containing the recurrent MH-based deletions among solid tumors. We agree that we should better explain this and we changed the legend accordingly:

pp. 35 Line 735

“a b, Number of samples carrying somatic truncating mutations and the position of the last amino acid (AA position) as identified in *ASXL1* gene across hematologic (a) and non-hematologic (b) tumors in COSMIC dataset. Truncating mutations among 1434 hematologic and 252 non-hematologic samples were analyzed. The proportions of MH-based deletion cases (orange) out of the total deletion cases (purple, green and orange) were compared between hematologic (153/376) and non-hematologic (4/103) tumors ($P < 0.00001$).”

As opposed to the figure that compared AMLs to solid tumors (which was removed), Figure 3 is describing absolute number of cases and not relative frequencies. We therefore consider chi-squared to be the appropriate statistical test in this analysis.

REVIEWERS' COMMENTS

Reviewer #1 (Remarks to the Author):

Thank you for addressing my comments. I have no additional comments.

Reviewer #3 (Remarks to the Author):

This was a very carefully considered and well-prepared rebuttal to my initial comments and concerns. I particularly appreciated the authors' engagement with the constructive intention of the comments and think they have done a very balanced revision which addresses the vast majority of my concerns. I commend the team for being so engaging with the review process and think the manuscript has benefitted enormously in terms of clarity/confidence of message and think the community will benefit as well. Well done.

Reviewer #4 (Remarks to the Author):

In this revision, authors performed new experiments and the manuscript is improved. Authors recapitulated MH-based deletions at ASXL1 and SRSF2 genes in four AML cell lines in addition to K562. This is a nice observation, which excludes the cell line specific artifact. However, it is still a major concern that the MMEJ pathway and the underlying mechanism involved in recurrent deletions in clonal hematopoiesis are not clearly addressed. Authors showed that PARP1 but not POLQ is involved and proposed a new POLQ-independent pathway responsible for preL-MMEJ deletions. PARP1 has additional roles in ssDNA break repair. The effect from aphidicolin treatment could be resulted from inhibition of DNA replication and is not necessarily caused by inhibition of DNA polymerases. With the current data, the mechanism of MMEJ underlying preL-MMEJ deletions is not clearly defined.

Response to reviewer 4.

We thank reviewer 4 for his comment. Accordingly, we have made the following corrections in the manuscript:

1. According to reviewer 4 comment regarding the aphidicolin inhibitory effect on DNA replication, in the revised abstract we added a statement suggesting that aphidicolin treatment can inhibit preL-MMEJ also through a general inhibition of replication and not necessarily through the direct inhibition of the replicative polymerases.

Please see revised abstract (pp. 2, line 35):

“In contrast, aphidicolin (an inhibitor of replicative polymerases and replication) treatment resulted in a significant reduction in preL-MMEJ. Altogether, our data indicate an association between polymerase theta independent MMEJ and clonal hematopoiesis and elucidate the mutational mechanisms involved in the very first steps of leukemia evolution.”

2. According to reviewer 4 concern that the MMEJ pathway and underlining mechanisms involved in recurrent deletions in clonal hematopoiesis are not clearly addressed, we have highlighted that the mechanisms responsible for preL-MMEJ are not fully resolved in this this manuscript, as can be reflected in the revised discussion (pp. 12, line 264):

“Furthermore, as the mechanism of MMEJ underlying preL-MMEJ deletions is not fully resolved in the current study, a more accurate description of the sub-pathway responsible for preL-MMEJ deletions is needed.”

3. As the editor and reviewer 4 suggested we share the opinion that the aphidicolin treatment could reduce the formation of preL-MMEJ as a general result of DNA replication inhibition and not directly through the inhibition of replicative polymerases.

The following sentence appears in the discussion (pp. 13, line 273):

“However, future studies are warranted to assess whether aphidicolin related reduction of preL-MMEJ is due to a direct inhibition of replicative polymerases or as a consequence of cell cycle arrest.”

4. As reviewer 4 correctly suggested, PARP1 has additional roles in ssDNA break repair. We added a sentence describing the different roles of PARP1 in DNA repair, stating that we could not rule out the involvement of such pathways in the formation of preL-MMEJ deletions in humans.

Please see the revised discussion (pp. 13, line 268):

“While PARP1 is known to regulate the MMEJ pathway³² it also plays a role in c-NHEJ¹⁷ and single strand break (SSB) repair³⁵. We cannot rule out that human preL-MMEJ might be the result of SSB.”